# MYC regulates ribosome biogenesis and mitochondrial gene expression programs through its interaction with host cell factor–1

Tessa M Popay[1], Jing Wang[2,3], Clare M Adams[4], Gregory Caleb Howard[1], Simona G Codreanu[5,6], Stacy D Sherrod[5,6], John A McLean[5,6], Lance R Thomas[1†], Shelly L Lorey[1], Yuichi J Machida[7], April M Weissmiller[1‡], Christine M Eischen[4], Qi Liu[2,3], William P Tansey[1,8]*

[1]Department of Cell and Developmental Biology, Vanderbilt University School of Medicine, Nashville, United States; [2]Department of Biostatistics, Vanderbilt University Medical Center, Nashville, United States; [3]Center for Quantitative Sciences, Vanderbilt University Medical Center, Nashville, United States; [4]Department of Cancer Biology, Thomas Jefferson University, Philadelphia, United States; [5]Center for Innovative Technology (CIT), Vanderbilt University, Nashville, United States; [6]Department of Chemistry, Vanderbilt University, Nashville, United States; [7]Department of Oncology, Mayo Clinic, Rochester, United States; [8]Department of Biochemistry, Vanderbilt University School of Medicine, Nashville, United States

*For correspondence:
william.p.tansey@vanderbilt.edu

Present address: †Oncocyte Corporation, Nashville, United States; ‡Department of Biology, Middle Tennessee State University, Murfreesboro, United States

**Competing interests:** The authors declare that no competing interests exist.

**Abstract** The oncoprotein transcription factor MYC is a major driver of malignancy and a highly validated but challenging target for the development of anticancer therapies. Novel strategies to inhibit MYC may come from understanding the co-factors it uses to drive pro-tumorigenic gene expression programs, providing their role in MYC activity is understood. Here we interrogate how one MYC co-factor, host cell factor (HCF)–1, contributes to MYC activity in a human Burkitt lymphoma setting. We identify genes connected to mitochondrial function and ribosome biogenesis as direct MYC/HCF-1 targets and demonstrate how modulation of the MYC–HCF-1 interaction influences cell growth, metabolite profiles, global gene expression patterns, and tumor growth in vivo. This work defines HCF-1 as a critical MYC co-factor, places the MYC–HCF-1 interaction in biological context, and highlights HCF-1 as a focal point for development of novel anti-MYC therapies.

## Introduction

MYC oncogenes (c-, L-, and N-) encode a family of related transcription factors that are overexpressed in a majority of cancers and responsible for ~100,000 cancer-related deaths in the United States each year (*Schaub et al., 2018*). Capable of acting as both transcriptional activators and repressors, MYC proteins (hereafter 'MYC') dimerize with their obligate partner MAX (*Blackwood and Eisenman, 1991*) to bind and regulate the expression of thousands of genes connected to the cell cycle, protein synthesis, metabolism, genome stability, apoptosis, and angiogenesis (*Tansey, 2014*). Fueled by reports that experimental inactivation of MYC promotes tumor regression in mice (*Alimova et al., 2019*; *Beaulieu et al., 2019*; *Giuriato et al., 2006*; *Jain, 2002*; *Soucek et al., 2013*), there is considerable interest in the idea that MYC inhibitors could form the

**eLife digest** Tumours form when cells lose control of their growth. Usually, cells produce signals that control how much and how often they divide. But if these signals become faulty, cells may grow too quickly or multiply too often. For example, a group of proteins known as MYC proteins activate growth genes in a cell, but too much of these proteins causes cells to grow uncontrollably.

With one third of all cancer deaths linked to excess MYC proteins, these molecules could be key targets for anti-cancer drugs. However, current treatments fail to target these proteins. One option for treating cancers linked to MYC proteins could be to target proteins that work alongside MYC proteins, such as the protein HCF-1, which can attach to MYC proteins.

To test if HCF-1 could be a potential drug target, Popay et al. first studied how HCF-1 and MYC proteins interacted using specific cancer cells grown in the laboratory. This revealed that when the two proteins connected, they activated genes that trigger rapid cell growth. When these cancer cells were then injected into mice, tumours quickly grew. However, when the MYC and HCF-1 attachments in the cancer cells were disrupted, the tumours shrank. This suggests that if anti-cancer drugs were able to target HCF-1 proteins, they could potentially reduce or even reverse the growth of tumours.

While further research is needed to identify drug candidates, these findings reveal a promising target for treating tumours that stem from over-abundant MYC proteins.

basis of broadly effective anticancer therapies. MYC itself, however, is widely viewed as undruggable (*Dang et al., 2017*), meaning that effective strategies to pharmacologically inhibit MYC will most likely come from targeting the co-factors with which it interacts to drive and sustain the malignant state (*Brockmann et al., 2013*; *Bryan et al., 2020*).

The interactome of MYC has been extensively interrogated (reviewed by *Baluapuri et al., 2020*). One effective strategy for prioritizing which of these interaction partners to study has been to focus on those that interact with conserved segments of the MYC protein, which are referred to as 'MYC boxes' (Mb) (*Meyer and Penn, 2008*). In addition to the highly conserved basic helix-loop-helix domain that interacts with MAX, six evolutionarily conserved MYC boxes have been described (*Baluapuri et al., 2020*). On average, MYC boxes are around 15 amino acid residues in length, and although it is clear that they each mediate multiple protein–protein interactions (*Kalkat et al., 2018*), a number of predominant interactors have been described for most of these segments: Mb0, for example, interacts with the general transcription factor TFIIF to stimulate transcription (*Kalkat et al., 2018*), MbI interacts with the ubiquitin ligase $SCF^{FBW7}$ to control MYC protein stability (*Welcker et al., 2004*), MbII interacts with the STAGA component TRRAP (*McMahon et al., 1998*) to regulate histone acetylation (*Kalkat et al., 2018*), and MbIIIb interacts with the chromatin-associated protein WDR5 (*Thomas et al., 2015*) to facilitate its recruitment to ribosomal protein genes (*Thomas et al., 2019*). The two remaining MYC boxes are less well understood, but MbIIIa is important for tumorigenesis (*Herbst et al., 2005*) and recruitment of HDAC3 to chromatin (*Kurland and Tansey, 2008*), and MbIV interacts with the ubiquitous chromatin-associated protein host cell factor (HCF)–1 (*Thomas et al., 2016*).

HCF-1 is an essential nuclear protein (*Goto et al., 1997*) that is synthesized as a 2035 amino acid precursor and proteolytically cleaved by O-GlcNAc transferase (OGT) (*Capotosti et al., 2011*) into a collection of amino- ($HCF-1_N$) and carboxy- ($HCF-1_C$) terminal fragments that remain associated. HCF-1 was first identified through its ability to assemble into a multiprotein–DNA complex with the herpes simplex virus transactivator VP16, but was later shown to function in uninfected cells as a co-factor for cellular transcription factors, and as part of the Sin3 and MLL/SET histone modifying complexes (*Wysocka and Herr, 2003*). The interaction of HCF-1 with VP16 is likely direct and is mediated by a tetrapeptide 'EHAY' motif in VP16 that binds to a region within the $HCF-1_N$ fragment known as the VP16-induced complex (VIC) domain (*Freiman and Herr, 1997*). This four residue HCF-1-binding motif (HBM)—consensus (D/E)-H-x-Y—is present in other viral and cellular transcription factors that interact directly with the HCF-1 VIC domain, including key cell cycle regulators such as the E2F family of proteins (*Tyagi et al., 2007*).

We identified HCF-1 as a MYC-associated protein through proteomic approaches, and demonstrated that the interaction occurs through the VIC domain of HCF-1 and an atypical HBM within MbIV that carries the sequence 'QHNY' (*Thomas et al., 2016*). Mutation of these four HBM residues to alanine disrupts the interaction of MYC with HCF-1 in vitro and reduces the ability of MYC to promote murine fibroblast tumor growth in nude mice (*Thomas et al., 2016*).

The small and well-defined interaction point between MYC and HCF-1, and the importance of this interaction to tumorigenesis, raise the possibility that the MYC–HCF-1 nexus could be a viable venue for discovery of novel anti-MYC therapies. If that venue is to be pursued, however, we need to place this interaction in biological context, identify the gene networks that are under its control, and determine whether the MYC–HCF-1 interaction is required for tumor initiation, maintenance, or both. Here we use a combination of loss- and gain-of function approaches to interrogate the role of the MYC–HCF-1 interaction in the context of a canonically MYC-driven cancer—Burkitt lymphoma. We demonstrate that the interaction between MYC and HCF-1 is directly involved in controlling the expression of genes linked to ribosome biogenesis, translation, and mitochondrial function. We define the impact of modulation of this interaction on cell growth, metabolism, and global gene expression patterns. And we show that disrupting the MYC–HCF-1 interaction promotes rapid and persistent tumor regression in vivo. This work reveals how MYC executes a core arm of its pro-tumorigenic gene expression changes, defines HCF-1 as a tumor-critical MYC co-factor, and provides proof-of-concept for a new way to inhibit MYC in the clinic.

## Results

### Bidirectional modulation of the MYC−HCF-1 interaction

To understand the role of HCF-1 in MYC function, we sought to use separation-of-function mutations in MYC that modulate interaction with HCF-1 in a predictable way. We therefore introduced a number of mutations in the atypical HBM of MYC (QHNY) that we expected to decrease—or increase—interaction with HCF-1, based on properties of prototypical HBM sequences (*Freiman and Herr, 1997*; *Figure 1A*). We substituted the MYC HBM for the canonical HBM from VP16 (VP16 HBM); we also mutated the invariant histidine of the HBM to glycine in the MYC (H307G) and VP16 (VP16 HBM:H307G) contexts, or we changed all four HBM residues to alanine (4A). A mutation in the separate WDR5-binding motif (WBM; *Thomas et al., 2015*) was our specificity control. We transiently expressed these full-length FLAG-tagged MYC proteins in 293T cells and measured their ability to interact with endogenous HCF-1 in a co-immunoprecipitation (co-IP) assay (*Figure 1B*). As expected, the 4A mutation disrupts the MYC–HCF-1 interaction, as do both histidine to glycine substitutions—confirming the essentiality of this core HBM residue to the MYC–HCF-1 association. In contrast, replacing the MYC HBM with the canonical VP16 sequence increases the amount of HCF-1 recovered in the co-IP. The enhanced binding of the VP16 HBM MYC mutant to HCF-1 is also observed in vitro using purified recombinant MYC (*Figure 1—figure supplement 1A*) and in vitro translated HCF-1$_{VIC}$ (*Figure 1—figure supplement 1B*). Based on these data, we conclude that the MYC HBM is an authentic HBM, and that its variation from the canonical HBM sequence leads to a tempered interaction with HCF-1. We also conclude that we can use the 4A and VP16 HBM mutations to probe the significance of the MYC–HCF-1 interaction through both loss- and gain-of-function approaches.

### The MYC–HCF-1 interaction stimulates proliferation of Burkitt lymphoma cells

To understand the cellular consequences of modulating the MYC–HCF-1 interaction, we engineered a system that allows us to express the 4A or VP16 HBM mutant MYC proteins as the sole form of MYC in a cell. We chose Ramos cells, a Burkitt lymphoma (BL)-derived line in which a *t(8;14)* translocation places one *MYC* allele under regulatory control of the immunoglobulin heavy chain enhancer (*Figure 1—figure supplement 1C*; *Wiman et al., 1984*). The untranslated *MYC* allele is not expressed in these cells (*Bemark and Neuberger, 2000*). Because sequences encoding the MYC HBM are contained within exon 3, we used CRISPR/Cas9-triggered homologous recombination of the translocated *MYC* allele to integrate an exon 3 switchable cassette for wild-type (WT) MYC, 4A, or VP16 HBM mutants, and confirmed appropriate integration by Southern blotting (*Figure 1—*

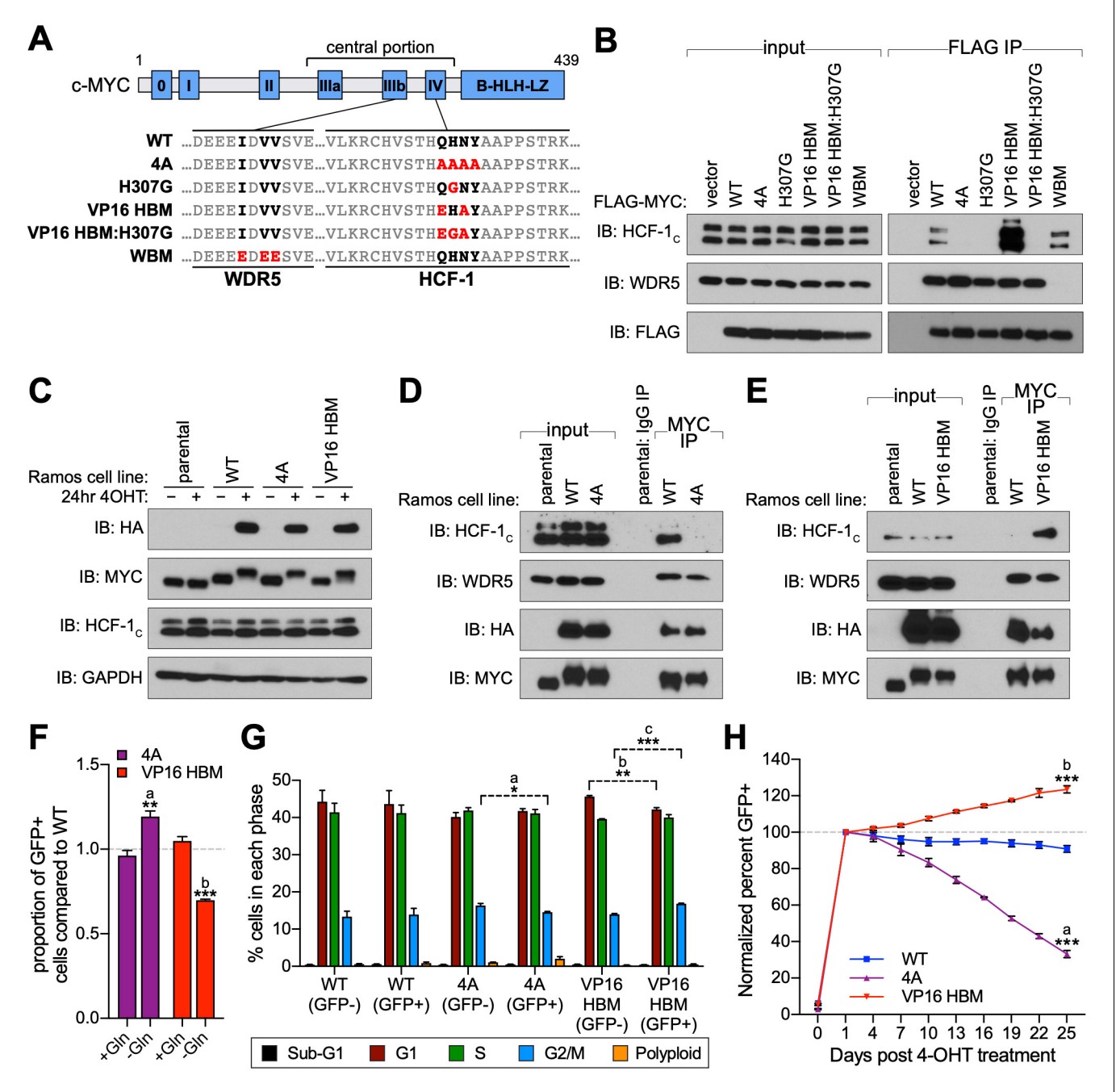

**Figure 1.** A gain- and loss-of-function system to study the MYC–host cell factor (HCF)–1 interaction. (**A**) Schematic of MYC, depicting the location of the six MYC boxes (Mb0–MbIV). MbIIIb carries a WDR5-binding motif (WBM). MbIV contains an HCF-1-binding motif (HBM). Residues relevant to the WBM or HBM are in bold, and residues mutated in this study are in red. (**B**) FLAG-tagged full-length MYC proteins carrying the mutations described in (**A**) were transiently expressed in 293T cells, recovered by anti-FLAG immunoprecipitation (IP), and the input, or IP eluates, probed for the presence of HCF-1$_C$, WDR5, or FLAG-tagged proteins by western blotting. (**C**) Western blot of lysates from parental (CRE-ER$^{T2}$) or switchable Ramos cells (wild-type [WT], 4A, or VP16 HBM) ±20 nM 4-OHT for 24 hr. Blots were probed with antibodies against the HA tag, MYC, HCF-1$_C$, and GAPDH. (**D** and **E**) Parental or switchable Ramos cells (WT, 4A, or VP16 HBM) were treated with 20 nM 4-OHT for 24 hr, lysates prepared, and IP performed using anti-IgG or anti-MYC antibodies. Input lysates and IP eluates were probed using antibodies against HCF-1$_C$, WDR5, HA tag, and MYC by western blotting. All lines in these experiments express CRE-ER$^{T2}$. (**F**) Switchable Ramos cell lines were pulsed with 20 nM 4-OHT for 2 hr to switch ~50% of cells, propagated for 3 days, and grown for 16 hr in media with or without glutamine. The impact of glutamine deprivation was measured by flow cytometry to determine the proportion of green fluorescent protein (GFP)-positive (switched) cells. For each of the mutants, the proportion of GFP-positive cells was normalized to that for WT cells. Shown are the mean and standard error for three biological replicates. Student's t-test between +Gln and −Gln

*Figure 1 continued on next page*

*Figure 1 continued*

was used to calculate p-values; a = 0.0066, b = 0.0002. (**G**) Switchable Ramos cells were pulsed with 4-OHT as in (**F**), grown for 7 days, and cell cycle distribution determined by propidium iodide (PI) staining and flow cytometry, binning cells according to whether they expressed GFP (GFP+, switched) or not (GFP−, unswitched). Shown are the mean and standard error for three biological replicates. Student's t-test between GFP− and GFP+ cells was used to calculate p-values; a = 0.033, b = 0.0041, c = 0.0006. (**H**) Switchable Ramos cells were pulsed with 4-OHT as in (**F**), and the proportion of GFP-positive cells measured by flow cytometry 24 hr after treatment and every 3 days following. For each of the replicates, the proportion of GFP-positive cells is normalized to that on day 1. Shown are the mean and standard error for three biological replicates. Student's t-test between WT and each of the mutants at day 25 was used to calculate p-values; a = 0.000028, b = 0.00026.

The online version of this article includes the following source data and figure supplement(s) for figure 1:

**Source data 1.** Raw data for MYC mutant growth curves.
**Figure supplement 1.** Validation of MYC mutants and switchable Ramos cell lines.
**Figure supplement 2.** Localization of MYC mutants and their impact on cell doubling time.

*figure supplement 1D and E*). In cells expressing an inducible CRE-ER$^{T2}$ recombinase, treatment with 4-hydroxytamoxifen (4-OHT) results in the excision of exon 3 of *MYC*, bringing in place a modified exon 3 that carries an HA-epitope tag and drives expression of P2A-linked green fluorescent protein (GFP). Twenty-four hours after 4-OHT treatment, at least 85% of cells in each population are switched—as monitored by GFP expression (*Figure 1—figure supplement 1F*), and we observe the expected appearance of HA-tagged MYC proteins, which migrate more slowly due to the presence of the epitope tag (*Figure 1C*). Importantly, the exchanged MYC proteins are expressed at levels comparable to endogenous MYC (*Figure 1C*), are predominantly nuclear (*Figure 1—figure supplement 2A and B*), and behave as expected, with the 4A mutant showing reduced (*Figure 1D*), and the VP16 HBM mutant enhanced (*Figure 1E*), interaction with endogenous HCF-1. Also as expected, these mutations have minimal impact on the interaction of MYC with WDR5. Thus, we successfully generated a system for inducible, selective, and bidirectional modulation of the MYC–HCF-1 interaction in the context of an archetypal MYC-driven cancer cell line.

To monitor the contribution of the MYC–HCF-1 interaction to cell proliferation, we pulsed each of our engineered Ramos lines with 4-OHT for 2 hr to generate approximately equally mixed populations of switched and unswitched cells. Based on the ability of MYC to drive glutamine addiction (*Jeong et al., 2014*) and cell cycle progression (*Pajic et al., 2000*), we monitored how the GFP-positive switched cells in the population compared to their unswitched counterparts in terms of glutamine-dependency (*Figure 1F*), cell cycle profiles (*Figure 1G*), and proliferation (*Figure 1H* and *Figure 1—source data 1*). We see that 4A switched cells have a selective advantage over the WT switch in their ability to grow without exogenous glutamine (*Figure 1F*). This advantage is likely due to loss of the MYC–HCF-1 interaction, as the VP16 HBM mutant cells have a corresponding deficit in growth under glutamine-starvation conditions (*Figure 1F*). When assayed in media replete with glutamine, cell cycle profiles for the two mutants are modestly altered compared to their WT counterparts, including small but statistically significant changes in the proportion of cells in G$_2$/M (*Figure 1G*), which again trend in opposite directions for the two MYC mutants—decreasing for the 4A-expressing cells and increasing for those that express the VP16 HBM mutant (*Figure 1G*). Finally, in long-term growth assays in complete media, we observe that 4A mutant cells are gradually lost from the culture over time, whereas there is a significant enrichment of VP16 HBM cells, compared to the WT control (*Figure 1H* and *Figure 1—source data 1*). The differences in representation of the two MYC mutants in these populations is unlikely due to apoptosis—we observe no differences in the proportion of sub-G$_1$ cells between the different switches (*Figure 1G*)—but tracks with changes in cell doubling time (*Figure 1—figure supplement 2C and D* and *Figure 1—source data 1*), which are increased for the 4A, and decreased for the VP16 HBM mutant cells. The altered and opposing impact of the 4A and VP16 HBM mutations in these assays leads us to conclude that the MYC–HCF-1 interaction promotes the glutamine-dependency—and rapid proliferative status—of these BL cells in culture.

## The MYC–HCF-1 interaction influences intracellular amino acid levels

As part of our survey of the impact of the MYC–HCF-1 interaction on Ramos cell processes, and because of its influence on glutamine dependency, we determined whether metabolite levels are altered in response to expression of the 4A or VP16 HBM MYC mutants. We performed global,

untargeted, mass spectrometry-based metabolomics on switched cells using reverse-phase liquid chromatography (RPLC) and hydrophilic interaction liquid chromatography (HILIC) separation methods. We detected ~2000 metabolites with each approach (*Figure 2A–F*), and there is strong consistency among biological replicates (*Figure 2—figure supplement 1A and B*). In general, more metabolites are significantly changed, and with a greater magnitude, for the 4A than the VP16 HBM MYC mutant (*Figure 2A–B and D–E* and *Figure 2—source data 1* and *2*). For both mutants, significantly changed metabolites group into a variety of categories, with a particular enrichment for those related to amino acid and lipids (*Figure 2C and F*). Comparing the direction of individual metabolite changes for the 4A and VP16 HBM mutants (*Figure 2—figure supplement 1C*), we note that a significant portion of the metabolite changes detected by both the RPLC and HILIC methods are in the same direction for the two MYC mutants. In general, these shared metabolite changes fail to cluster strongly into biological pathways; the only significantly enrichment being glycerophospholipid metabolism (*Figure 2—figure supplement 1D*). Focusing on metabolite changes that occur in opposite directions for the 4A and VP16 HBM mutants, however, we observe significant enrichment in pathways linked to nitrogen and amino acid metabolism (*Figure 2G*). There is a clear anti-correlation between the impact of the 4A and VP16 HBM mutations on metabolites connected to aspartic acid (*Figure 2H*), and we observe that intracellular levels of glutamine (and associated metabolites) are increased in the 4A and decreased in the VP16 HBM mutant cells (*Figure 2—figure supplement 1E and F*). Notably, these changes in intracellular amino acid levels are not confined to aspartic acid and glutamine, but there is a general tendency for amino acid levels to be increased in 4A and decreased in VP16 HBM mutant cells, compared to the WT switch (*Figure 2—figure supplement 1C* and *Table 1*). Based on these data, we conclude that the MYC–HCF-1 interaction, directly or indirectly, plays a global role in influencing intracellular amino acid levels in this setting.

## The MYC–HCF-1 interaction influences expression of genes connected to ribosome biogenesis and the mitochondrial matrix

Next, we used RNA-sequencing (RNA-Seq) to monitor transcriptomic changes associated with modulating the MYC–HCF-1 interaction. Twenty-four hours after switching, we observed changes in the levels of ~4000 transcripts in the 4A, and ~3600 transcripts in the VP16 HBM, cells compared to the WT switch (*Figure 3—figure supplement 1A* and *Figure 3—source data 1* and *2*). These changes are highly consistent among biological replicates (*Figure 3—figure supplement 1B*) and modest in magnitude (*Figure 3A*), congruous with what is typically reported for MYC (*Levens, 2002*; *Nie et al., 2012*). We confirmed for a representative set of genes that these changes are dependent on switching (*Figure 3—figure supplement 1C*). Gene ontology (GO) enrichment analysis revealed that transcripts decreased by introduction of the 4A mutant are strongly linked to ribosome biogenesis, tRNA metabolism, and the mitochondrial matrix (*Figure 3B*), while those that are induced have links to transcription, cholesterol biosynthesis, and chromatin. For the VP16 HBM MYC mutant, decreased transcripts cluster in categories mostly related to the centrosome and the cell cycle (*Figure 3C*). What is particularly striking, however, is that transcripts that are induced by the VP16 HBM protein have a pattern of clustering that is almost the exact opposite of those suppressed by the 4A mutant—including ribosome biogenesis, tRNA metabolism, and the mitochondrial matrix.

If anti-correlations between these gain- and loss-of-function mutants can be used to reveal MYC–HCF-1 co-regulated processes, the above data highlight protein synthesis and mitochondrial function as key points of convergence for the interaction of MYC with HCF-1. To explore this on a gene-by-gene basis, we compared individual gene expression changes that were either the same, or opposite, in direction for the 4A and VP16 HBM mutants (*Figure 3D*, *Figure 3—figure supplement 1D* and *Figure 3—source data 3*). Transcripts decreased by both mutations show modest enrichment in categories connected to immune signaling and cell adhesion (*Figure 3E*, left), whereas increased transcripts are primarily enriched in those encoding histones (*Figure 3E*, right). Turning to transcripts that change in opposite directions with each mutant, those that are induced by the 4A mutant are moderately enriched in categories relating to kinase function and the cell cycle (*Figure 3F*, left), while those that are reduced by the 4A mutant are strongly enriched in categories connected to ribosome biogenesis and the mitochondrial matrix (*Figure 3F*, right). The genes represented in each of these categories are shown in *Figure 3—figure supplement 2A and B*. This analysis confirms that reciprocal changes we observed for the GO categories in *Figure 3B and C* results from reciprocal changes in the expression of a common set of genes. From our data, we conclude

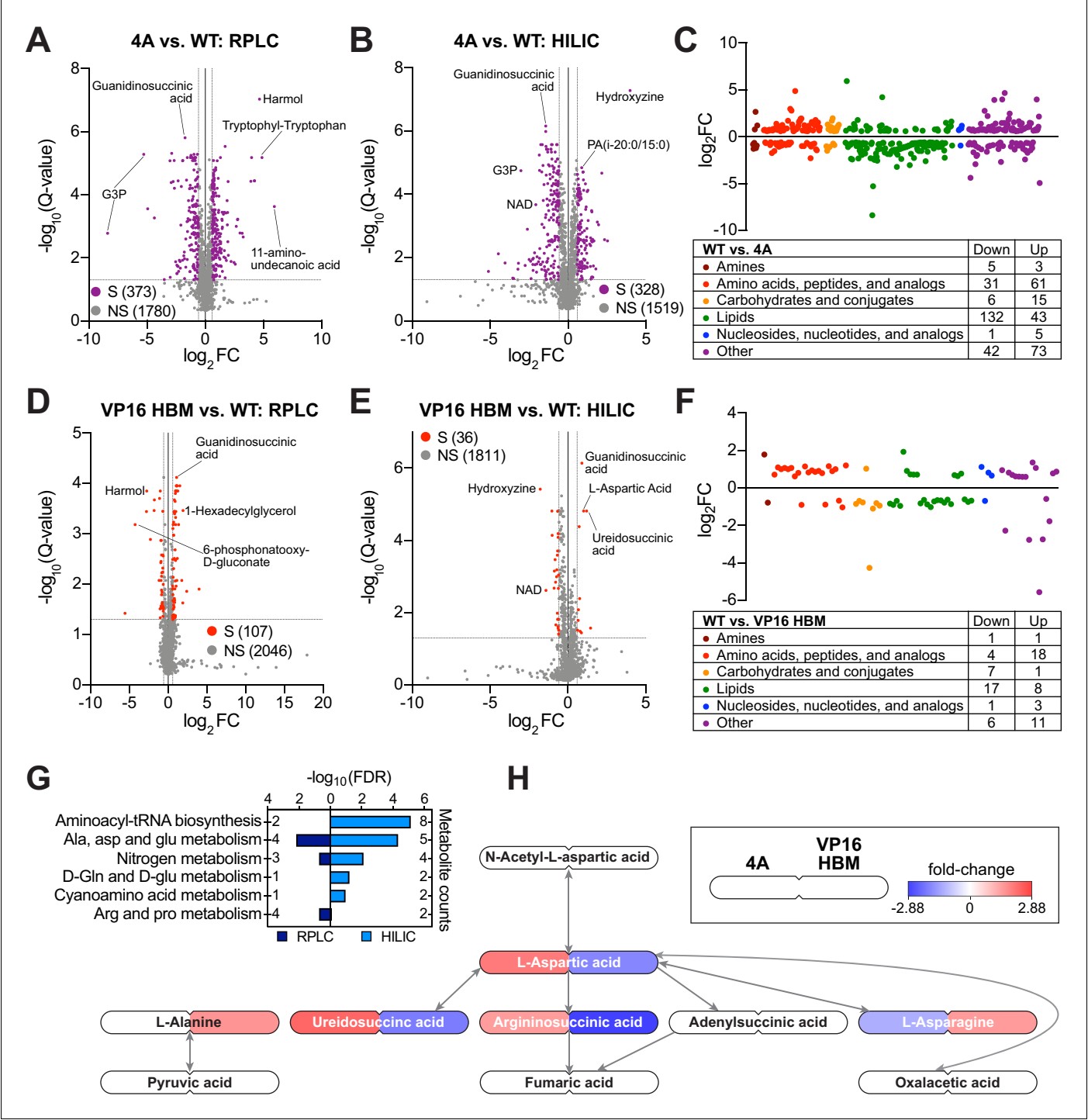

**Figure 2.** The MYC–host cell factor (HCF)–1 interaction influences intracellular amino acid levels. (**A** and **B**) Volcano plots of metabolites detected by reverse-phase liquid chromatography (RPLC) (**A**) or hydrophilic interaction liquid chromatography (HILIC) (**B**) in wild-type (WT) and 4A switchable Ramos cells treated for 24 hr with 20 nM 4-OHT. Metabolites that were significantly (S) changed (false discovery rate [FDR] < 0.05 and |FC| > 1.5) with the 4A MYC mutant compared to WT are colored. Non-significant (NS) changes are in gray. Five biological replicates for WT and four biological replicates for 4A were used to calculate Q-values and fold-changes (FCs). Select metabolites are indicated. (**C**) Classification of metabolites that were significantly changed (FDR < 0.05 and |FC| > 1.5) with the 4A mutant compared to WT cells. (**D** and **E**) Volcano plots of metabolites detected by RPLC (**D**) or HILIC (**E**) in WT and VP16 HCF-1-binding motif (HBM) switchable Ramos cells treated for 24 hr with 20 nM 4-OHT. Metabolites that were significantly (S) changed (FDR < 0.05 and |FC| > 1.5) with the VP16 HBM MYC mutant compared to WT are colored. Non-significant (NS) changes are in gray. Five biological replicates for WT and VP16 HBM were used to calculate Q-values and FCs. Select metabolites are indicated. (**F**) Classification of metabolites

*Figure 2 continued on next page*

**Figure 2 continued**

that were significantly changed (FDR < 0.05 and |FC| > 1.5) with the VP16 HBM mutant compared to WT cells. **(G)** Annotated metabolites from **Figure 2—figure supplement 1C** that were changed in the opposite direction for the 4A and VP16 HBM mutants were independently subjected to pathway enrichment analysis. Pathways with FDR < 0.2 for either RPLC and HILIC are shown. **(H)** Metabolites (FDR < 0.05) in the 'alanine, aspartate, and glutamate metabolism' pathway that were impacted by the 4A (left) and VP16 HBM (right) MYC mutants. Node color represents the FC over WT. The remainder of the pathway is shown in **Figure 2—figure supplement 1E and F**.

The online version of this article includes the following source data and figure supplement(s) for figure 2:

**Source data 1.** Ramos 4A and VP16 host cell factor (HCF)–1-binding motif reverse-phase liquid chromatography significant changes.
**Source data 2.** Ramos 4A and VP16 host cell factor (HCF)–1-binding motif hydrophilic interaction liquid chromatography significant changes.
**Figure supplement 1.** Impact of the 4A and VP16 HCF-1-binding motif (HBM) mutants on Ramos cell metabolites.

that the MYC–HCF-1 interaction plays an important role in influencing the expression of genes that promote ribosome biogenesis and maintain mitochondrial function.

Finally, we interrogated our RNA-Seq data set for transcript changes that correlate with the widespread changes in amino acid levels that occur upon modulation of the MYC–HCF-1 interaction. Here we discovered that the accumulation of amino acids we observe with the 4A mutant is generally matched with a decrease in transcripts of cognate aminoacyl-tRNA synthetases (**Figure 3—**

**Table 1.** Loss of the MYC–host cell factor (HCF)–1 interaction promotes amino acid accumulation. Modulating the MYC–HCF-1 interaction alters intracellular amino acid levels. All data are derived from switchable Ramos cells treated with 20 nM 4-OHT for 24 hr. Amino acid levels were measured following separations by hydrophilic interaction liquid chromatography (HILIC). Q-values and fold-changes (FCs) were calculated between the mutants and wild-type (WT). Five biological replicates for WT and VP16 HCF-1-binding motif (HBM) and four biological replicates for 4A were analyzed. Q-value < 0.05 are highlighted in green, FC > 0 in red, and FC < 0 in blue. Confidence levels reflect the confidence in metabolite identification; L1 is validated, L2 is putative, and L3 is tentative. ND = not detected; NS = not significant.

| | 4A | | VP16 HBM | | Confidence level |
|---|---|---|---|---|---|
| | Q-value | FC | Q-value | FC | |
| Glycine | 2.52E-04 | 1.31 | 3.11E-05 | −1.48 | L3 |
| L-alanine | 9.97E-05 | 1.62 | NS | | L2 |
| L-arginine | 2.31E-02 | 1.18 | NS | | L1 |
| L-asparagine | 6.67E-05 | 1.5 | 2.02E-03 | −1.23 | L2 |
| L-aspartic acid | 1.24E-03 | −1.37 | 1.54E-05 | 2.01 | L2 |
| L-cysteine | ND | | ND | | |
| L-glutamic acid | 2.53E-03 | 1.46 | NS | | L2 |
| L-glutamine | 2.33E-05 | 1.46 | 9.38E-05 | −1.41 | L2 |
| L-histidine | 8.70E-05 | 1.64 | NS | | L2 |
| L-isoleucine | 1.10E-03 | 1.39 | 2.36E-02 | −1.21 | L2 |
| L-leucine | 8.63E-04 | 1.33 | 2.55E-02 | −1.18 | L2 |
| L-lysine | 1.66E-02 | 1.19 | NS | | L2 |
| L-methionine | 6.51E-05 | 1.61 | 4.39E-03 | −1.27 | L2 |
| L-phenylalanine | 4.71E-04 | 1.48 | 3.46E-02 | −1.22 | L2 |
| L-proline | 1.01E-02 | 1.17 | NS | | L2 |
| L-serine | 7.48E-05 | 1.59 | NS | | L1 |
| L-threonine | 9.63E-05 | 1.54 | 1.25E-02 | −1.17 | L2 |
| L-tryptophan | 1.56E-04 | 1.64 | NS | | L1 |
| L-tyrosine | 1.22E-04 | 1.59 | 1.9E-02 | −1.18 | L2 |
| L-valine | 4.41E-02 | 1.14 | NS | | L3 |

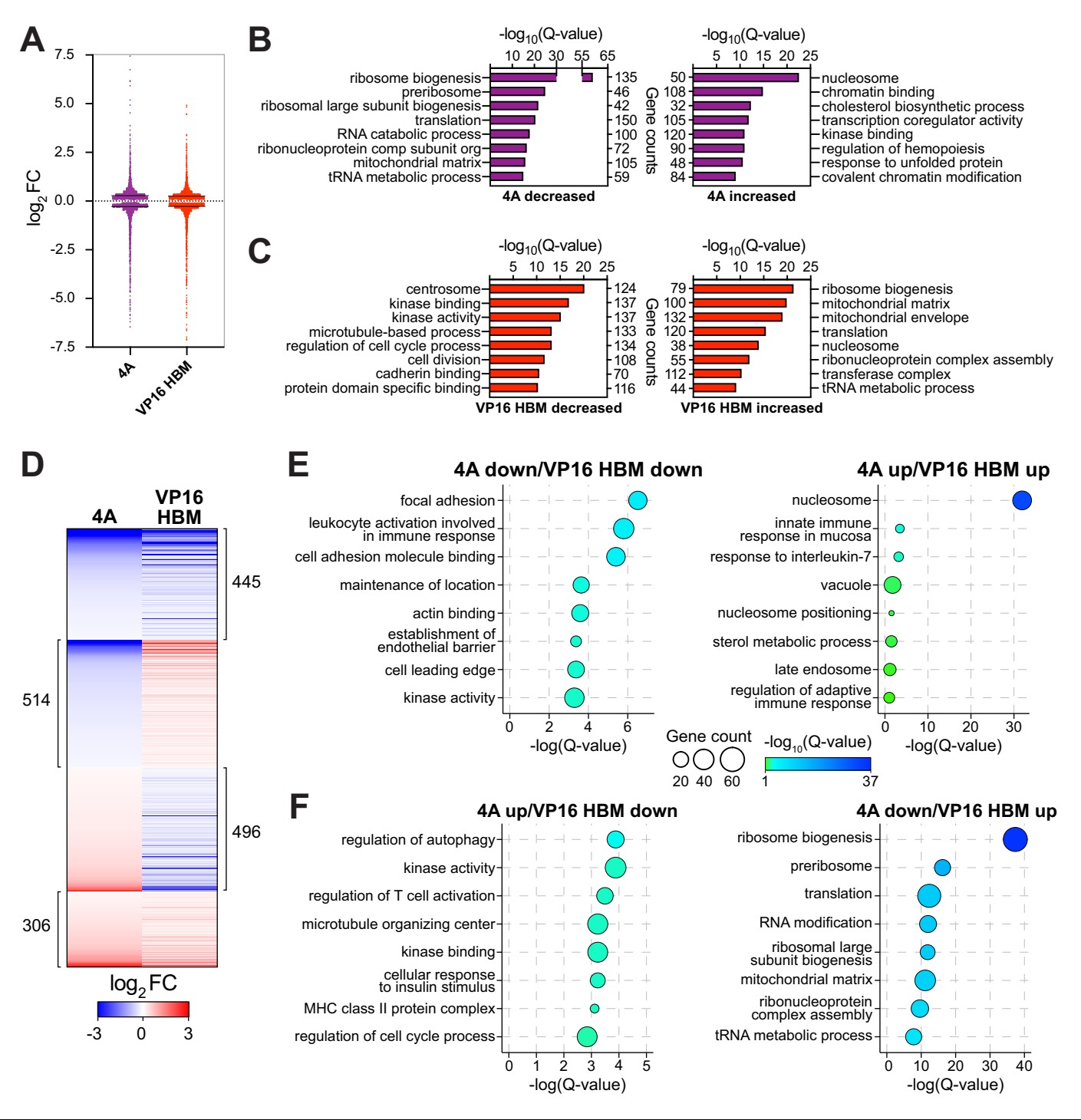

**Figure 3.** The MYC–host cell factor (HCF)–1 interaction influences the expression of genes involved in ribosome biogenesis and mitochondrial pathways. Switchable Ramos cells were treated with 20 nM 4-OHT for 24 hr, RNA isolated, and RNA-Seq performed. (A) Scatterplot showing the distribution of $\log_2$FC of significant (false discovery rate [FDR] < 0.05) RNA-Seq changes with the 4A and VP16 HCF-1-binding motif (HBM) MYC mutants, compared to the wild-type (WT) switch. Solid lines represent the median $\log_2$FC for decreased (4A: −0.2858; VP: −0.2747) and increased (4A: 0.281; VP: 0.2558) transcripts compared to WT. For clarity, some data points were excluded; these data points are highlighted in *Figure 3—source data 1* and *2*. (B and C) Categories from the top eight families in gene ontology (GO) enrichment analysis of significant (FDR < 0.05) gene expression changes under each condition (B: 4A; C: VP16 HBM). (D) Heatmap showing the $\log_2$FC of significantly (FDR < 0.05) changed transcripts that are altered in expression in both the 4A and VP16 HBM mutants. Transcripts are clustered according to the relationship in expression changes between the 4A and VP16 HBM mutants, and ranked by the $\log_2$FC for the 4A mutant. Scale of heatmap is limited to [−3,3]. (E and F) Gene clusters in (D) were subject to

*Figure 3 continued on next page*

*Figure 3 continued*

GO enrichment analysis, and the top eight categories are shown for the correlative (E) or anti-correlative (F) clusters. The Q-value of categories is represented by bubble color, and the number of genes present in a category is represented by bubble size. The genes in these categories are identified in *Figure 3—figure supplement 2A and B*.

The online version of this article includes the following source data and figure supplement(s) for figure 3:

**Source data 1.** Ramos 4A 24 hr RNA-Seq significant changes.
**Source data 2.** Ramos VP16 host cell factor (HCF)–1-binding motif 24 hr RNA-Seq significant changes.
**Source data 3.** Ramos 4A and VP16 host cell factor (HCF)–1-binding motif 24 hr RNA-Seq shared significant changes.
**Figure supplement 1.** Gene expression changes induced by the 4A and VP16 host cell factor (HCF)–1-binding motif (HBM) mutants.
**Figure supplement 2.** Gene ontology (GO) enrichment analysis of correlative and anti-correlative effects on gene expression.
**Figure supplement 3.** Amino acids and their cognate tRNA-ligases are influenced by the MYC–host cell factor (HCF)–1 interaction.

*figure supplement 3A*)—and vice versa for the decreased amino acid levels in the gain-of-function VP16 HBM mutant (*Figure 3—figure supplement 3B*). The reciprocal relationship of amino acid levels and tRNA ligase expression changes in response to the 4A and VP16 HBM mutants is consistent with the notion that defects in tRNA charging lead to compensatory changes in amino acid uptake (*Guan et al., 2014*; *Harding et al., 2000*), further reinforcing the concept that a key biological context in which MYC and HCF-1 function together is protein synthesis.

## Ribosome biogenesis and mitochondrial matrix genes respond rapidly to HCF-1 depletion

As a challenge to the concept that ribosome biogenesis and mitochondrial matrix genes are controlled via the MYC–HCF-1 interaction, we asked whether expression of these genes is impacted by acute depletion of HCF-1, mediated via the dTAG method (*Nabet et al., 2018*). We used CRISPR/Cas9-triggered homologous recombination to integrate an mCherry-P2A-FLAG-FKBP12$^{F36V}$ cassette into the *HCFC1* locus in Ramos cells; the effect of which is to amino-terminally tag HCF-1$_N$ with the FLAG epitope and FKBP12$^{F36V}$ tags, and to mark the population of modified cells by mCherry expression (*Figure 4—figure supplement 1A*). Because the *HCFC1* locus resides on the X-chromosome, and because Ramos cells are derived from an XY patient, only a single integration event is needed. Tagged cells sorted by fluorescence-activated cell sorting display the expected shift in apparent molecular weight of HCF-1$_N$ and the appearance of an appropriately-sized FLAG-tagged species (*Figure 4A*). Addition of the dTAG-47 degrader results in the rapid and selective disappearance of the HCF-1$_N$ fragment; the HCF-1$_C$ fragment is largely unaffected by up to 24 hr of dTAG-47 treatment (*Figure 4B*). Consistent with the known functions of HCF-1 (*Julien and Herr, 2003*), treated cells display altered cell cycle profiles (*Figure 4—figure supplement 1B*), but appear to be able to complete at least one round of cell division, as notable deficits in proliferation are only evident 48 hr after dTAG-47 addition (*Figure 4C* and *Figure 4—source data 1*). These data reveal that the HCF-1$_N$ fragment is essential in Ramos cells, and that early time point analyses should be resistant to complicating effects of HCF-1$_N$ degradation on cell proliferation.

We performed RNA-Seq analysis 3 hr after addition of dTAG-47—a time point at which the majority of HFC-1$_N$ is degraded (*Figure 4B*). Despite the early time point, we identified ~4500 significant transcript changes associated with dTAG-47 treatment of sorted cells (*Figure 4—figure supplement 1C* and *Figure 4—source data 2*). These changes are equally divided between increased and decreased, although decreased transcripts are generally more impacted (larger median fold-change [FC]) than those that are induced (*Figure 4D*). Seventy-five of these differentially expressed genes are also altered in response to dTAG-47 treatment of unmodified Ramos cells (*Figure 4—figure supplement 1D and E* and *Figure 4—source data 3*) and were excluded from further analyses. GO enrichment analysis showed that transcripts reduced by HCF-1$_N$ degradation are similar in kind to those reduced by the 4A mutation in MYC—including ribosome biogenesis and tRNA metabolic processes (*Figure 4E*)—while those induced by HCF-1$_N$ degradation tend to be cell cycle-connected (*Figure 4E* and *Figure 4—figure supplement 1F*). We validated representative transcript changes by reverse transcription and quantitative PCR (RT-qPCR; *Figure 4—figure supplement 1G*). Importantly, many of the genes that are differentially expressed upon HCF-1$_N$ degradation are also differentially expressed in the presence of either the 4A or VP16 HBM mutants (*Figure 4—figure supplement 1H*), and we identified a union set of ~450 genes—oppositely regulated by the 4A and

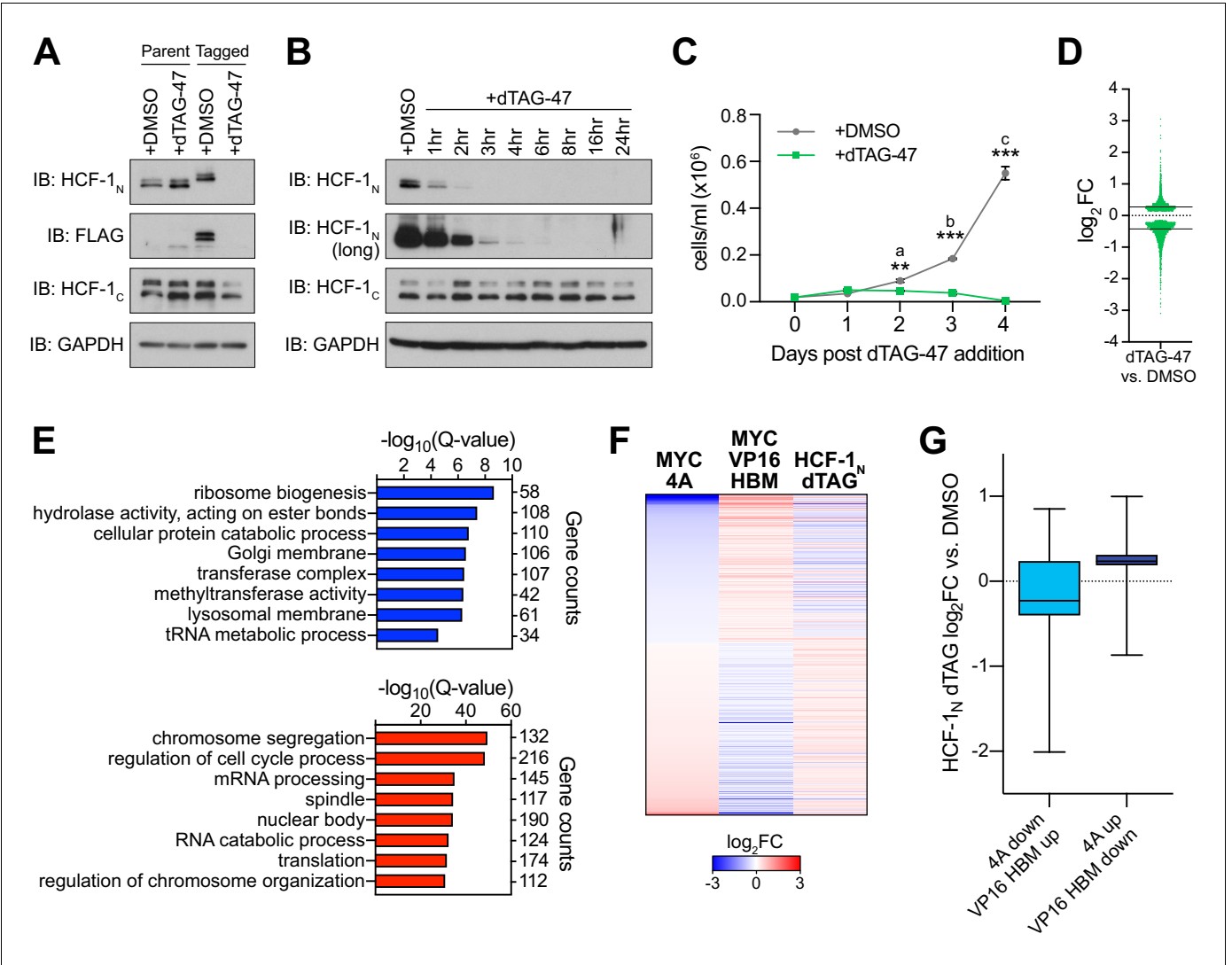

**Figure 4.** Genes regulated by the MYC−host cell factor (HCF)–1 interaction are impacted by loss of HCF-1. (**A**) Western blot, comparing the effects of treating untagged parental cells or FKBP$^{FV}$-HCF-1$_N$ Ramos cells with DMSO or 500 nM dTAG-47 for 24 hr. Blots for HCF-1$_N$, FLAG tag, HCF-1$_C$, and GAPDH are shown. (**B**) Western blot of lysates from FKBP$^{FV}$-HCF-1$_N$ Ramos cells treated with 500 nM dTAG-47 for varying times, compared to cells treated with DMSO for 24 hr. Shown are short and long exposures of HCF-1$_N$, and HCF-1$_C$, with a GAPDH loading control. (**C**) Growth curve of FKBP$^{FV}$-HCF-1$_N$ Ramos cells treated with DMSO or 500 nM dTAG-47. Cells were counted every 24 hr for 4 days after plating. Shown are the mean and standard error for three biological replicates. Student's t-test was used to calculate p-values; a = 0.0029, b = 0.000051, c = 0.000040. (**D**) Scatterplot showing the distribution of log$_2$FC in RNA-Seq comparing DMSO to 3 hr of 500 nM dTAG-47 treatment (degradation of HCF-1$_N$). Solid lines represent the median log$_2$FC for decreased (−0.425655) and increased (0.270428) transcripts. For clarity, one data point was excluded; this data point is highlighted in *Figure 4—source data 2*. (**E**) Gene ontology enrichment analysis of transcripts significantly (false discovery rate [FDR] < 0.05) decreased (top) and increased (bottom) in expression following treatment of FKBP$^{FV}$-HCF-1$_N$ Ramos cells with dTAG-47 for 3 hr. Excluded from this analysis are transcripts that were significantly changed when parental Ramos cells were treated with dTAG-47 for 3 hr. (**F**) Heatmap showing the log$_2$FC of transcripts with significantly (FDR < 0.05) changed expression, as measured by RNA-Seq, under the indicated conditions. Transcripts are clustered according to the relationship in expression changes between the 4A and VP16 HBM mutants, and ranked by the log$_2$FC for the 4A mutant. Scale of heatmap is limited to [−3,3]. (**G**) Box-and-whisker plot showing the relationship between transcripts that are anti-correlated between the 4A and VP16 HBM MYC mutants, and significantly changed with the degradation of HCF-1$_N$. Box denotes the 25th to 75th percentile, whiskers extend from minimum to maximum point, and middle line marks the median (4A down/VP16 HBM up: −0.2276; 4A up/VP16 HBM down: 0.2349).

The online version of this article includes the following source data and figure supplement(s) for figure 4:

**Source data 1.** Raw data for host cell factor (HCF)–1$_N$ degradation growth curve.
**Source data 2.** Ramos host cell factor (HCF)–1$_N$ degradation RNA-Seq significant changes.
**Source data 3.** Ramos untagged RNA-Seq significant changes.
**Figure supplement 1.** Inducible degradation of host cell factor (HCF)–1$_N$.

VP16 HBM mutants—the expression of which also changes when HCF-1$_N$ is destroyed (*Figure 4F*). Within this set, loss of HCF-1$_N$ tends to mimic the loss of interaction 4A mutant (*Figure 4G*). Moreover, within the cohort of transcripts that are reduced by both HCF-1$_N$ destruction and the 4A mutation, we see clear representation of genes connected to ribosome biogenesis and the mitochondrial matrix (*Figure 4—figure supplement 1I and J*). Although performing RNA-Seq at this early time likely underestimates the impact of loss of HCF-1$_N$ on the transcriptome, the presence of these ribosome biogenesis and mitochondrial matrix genes at the point of coalescence of all our RNA-Seq experiments strongly suggests that they are directly controlled by the MYC–HCF-1 interaction.

## Most HCF-1 binding sites on chromatin are bound by MYC

To help identify direct transcriptional targets of the MYC–HCF-1 interaction, we next compared the genomic locations bound by MYC and HCF-1$_N$ in Ramos cells. We performed ChIP-Seq using an antibody against the amino-terminus of HCF-1 (*Machida et al., 2009*), and identified ~1900 peaks for HCF-1$_N$ (*Figure 5—source data 1*), the majority of which are promoter proximal (*Figure 5A*). These peaks occur at genes enriched in functions connected to HCF-1 (*Minocha et al., 2019*), including the mitochondrial envelope, the cell cycle, as well as ribonucleoprotein complex biogenesis (*Figure 5B*). Known (*Figure 5C*) and de novo (*Figure 5—figure supplement 1A*) motif analysis revealed that HCF-1$_N$ peaks are enriched in DNA sequences linked to nuclear respiratory factor (NRF)–1, as well as the Sp1/Sp2 family of transcription factors. Although linked to NRF-1, the 'CATGCG' motif is also a non-canonical E-box that MYC binds in vitro and in vivo (*Blackwell et al., 1993*; *Haggerty et al., 2003*; *Morrish et al., 2003*; *Shi et al., 2014*). Overlaying these data with our previous ChIP-Seq analysis of MYC in Ramos cells (*Thomas et al., 2019*), we see that 85% of these HCF-1$_N$ peaks are also bound by MYC (*Figure 5D* and *Figure 5—source data 2*). The relationship between MYC and HCF-1$_N$ at these sites is intimate (*Figure 5E and F* and and *Figure 5—figure supplement 1B*), and sites of co-binding tend to have higher signals for MYC (*Figure 5G*) and HCF-1 (*Figure 5—figure supplement 1C*) than instances where each protein binds alone. As expected from the strong coalescence of MYC and HCF-1$_N$ binding events, the properties of shared MYC–HCF-1$_N$ peaks are very similar to those of HCF-1$_N$ alone, in terms of promoter-proximity (*Figure 5—figure supplement 1D*), GO enrichment categories (*Figure 5—figure supplement 1E*), and motif representation (*Figure 5—figure supplement 1F*). We conclude that most HCF-1$_N$ binding sites on chromatin in Ramos cells occur at promoter proximal sites and that the majority of these sites are also bound by MYC.

We previously reported that WDR5 has an important role in recruiting MYC to chromatin at a cohort of genes overtly linked to protein synthesis, including more than half of the ribosomal protein genes (*Thomas et al., 2019*). To determine whether these genes are also bound by HCF-1, we compared our HCF-1$_N$ and MYC ChIP-Seq data to those we generated for WDR5 in this setting. Interestingly, there is little overlap of binding sites for MYC, WDR5, and HCF-1$_N$ in Ramos cells, with just ~5% of MYC– HCF-1$_N$ co-bound sites also being bound by WDR5 (*Figure 5H*). Moreover, of the 88 sites bound by all three proteins, only three of these are sites where WDR5 has a functional role in MYC recruitment. Thus, despite the fact that both WDR5 and HCF-1 are often members of the same protein complex (*Cai et al., 2010*), and despite both of them having links to key aspects of protein synthesis gene expression, the two proteins associate with MYC at distinct and separate regions of the genome.

Finally, we overlaid the physical location of MYC and HCF-1$_N$ on chromatin with gene expression changes we had monitored in earlier experiments. Looking at genes displaying promoter proximal binding of MYC and HCF-1$_N$—where clear gene assignments can be made—we see that approximately one-third of these genes are differentially regulated in the presence of either the 4A or VP16 HBM MYC mutants, and that this is significantly more than that predicted by chance alone (*Figure 5I* and *Figure 5—figure supplement 1G*). For the 4A mutant, a slightly greater proportion of co-bound genes are downregulated, while the opposite is true for the VP16 HBM mutant (*Figure 5—figure supplement 1G*). A relatively small cohort of MYC–HCF-1$_N$ co-bound genes are oppositely impacted by the 4A and VP16 HBM mutants (*Figure 5—figure supplement 1G*; '4A/VP'), but comparing these with those deregulated by depletion of HCF-1$_N$ (*Figure 5J*), we again see that a majority are connected to ribosome biogenesis and mitochondria, and that most are positively regulated by HCF-1 and the MYC–HCF-1 interaction. Together, these data strongly support the notion

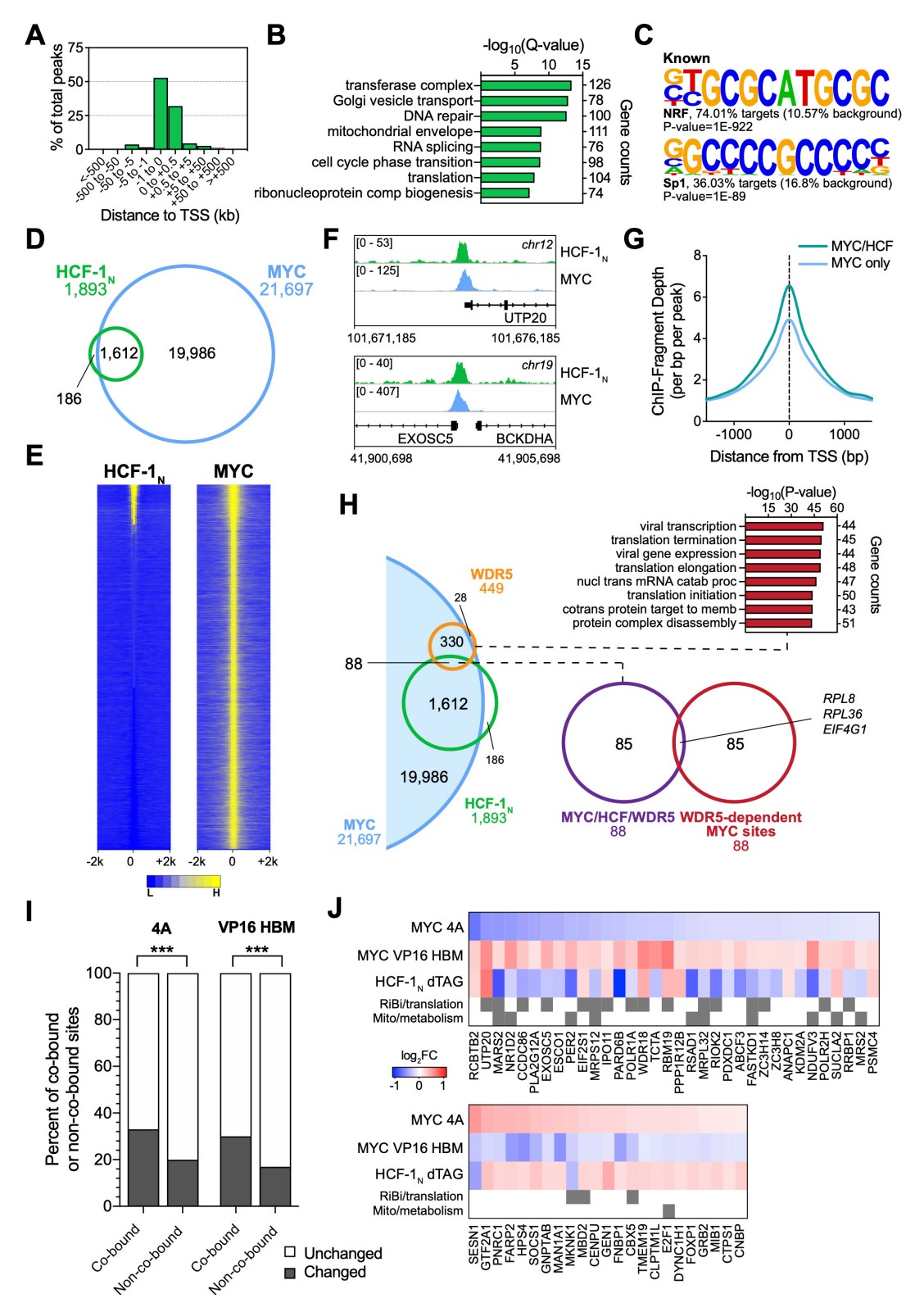

**Figure 5.** MYC is a widespread binding partner of host cell factor (HCF)–1 on chromatin. (**A**) Distribution of HCF-1$_N$ peaks in Ramos cells in relation to the nearest transcription start site (TSS), as determined by ChIP-Seq. (**B**) Gene ontology (GO) categories strongly represented in genes nearest to HCF-1$_N$ peaks in Ramos cells. (**C**) Known motif analysis of HCF-1$_N$ peaks in Ramos cells. Two of the most highly enriched motifs are shown, as well as the percentage of target and background sequences with the motif, and the p-value. (**D**) Venn diagram showing HCF-1$_N$ and MYC peaks in Ramos cells,

*Figure 5 continued on next page*

*Figure 5 continued*

and the number of regions that overlap between the data sets. ChIP-Seq data for MYC are from GSE126207. (E) Heatmap of all MYC peaks in Ramos cells (from GSE126207) and the corresponding region in Ramos HCF-1$_N$ ChIP-Seq, representing the combined average of normalized peak intensity in ±2 kb regions surrounding the peak centers with 100 bp bin sizes. Ranking is by peak intensity in HCF-1$_N$. (F) Example Integrative Genomics Viewer (IGV) screenshots of regions that have overlapping peaks for MYC and HCF-1$_N$ in Ramos cells. (G) Normalized MYC ChIP-Seq fragment counts where peaks overlap with HCF-1$_N$ (MYC/HCF), compared to where they do not overlap (MYC) in Ramos cells. Data are smoothed with a cubic spline transformation. (H) Venn diagram showing relationship between HCF-1$_N$, MYC, and WDR5 peaks in Ramos cells, and the overlap between co-bound genes and genes for which WDR5 is responsible for MYC recruitment. GO enrichment analysis of genes co-bound by MYC and WDR5—taken from *Thomas et al., 2019*—is also shown. (I) The proportion of protein-coding genes that were co-bound by promoter-proximal MYC and HCF-1$_N$ by ChIP-Seq or an equal number of non-co-bound genes were compared to transcripts that were unchanged or significantly changed (false discovery rate [FDR] < 0.05) with the 4A and VP16 HCF-1-binding motif (HBM) mutants by RNA-seq. p-Value for the 4A mutant is $1.982 \times 10^{-14}$ and $6.933 \times 10^{-14}$ for the VP16 HBM mutant. (J) Heatmap showing genes that are co-bound by promoter proximal MYC and HCF-1$_N$ in Ramos cells, have anti-correlative gene expression changes between for the 4A and VP16 MYC mutants, and have significant gene expression changes with HCF-1$_N$ degradation. Genes that fall into GO categories relating to ribosome biogenesis or translation (RiBi/translation), and mitochondrial function or metabolism (Mito/metabolism) are highlighted.

The online version of this article includes the following source data and figure supplement(s) for figure 5:

Source data 1. Ramos host cell factor (HCF)–1$_N$ annotated ChIP-Seq peaks.

Source data 2. Annotated intersect of ChIP-Seq peaks for host cell factor (HCF)–1$_N$ and MYC in Ramos cells.

Figure supplement 1. Comparison of MYC and host cell factor (HCF)–1$_N$ localization on Ramos cell chromatin.

that ribosome biogenesis and mitochondrially connected genes are direct targets of the MYC–HCF-1 interaction.

## MYC and HCF-1 bind chromatin independent of their ability to interact

It has been reported that deletion of MbIV from N-MYC reduces the ability of MYC:MAX dimers to bind DNA (*Cowling et al., 2006*). This phenotype is unrelated to the MYC HBM, however, as we determined that neither the 4A nor the VP16 HBM mutations have an overt impact on the binding of recombinant MYC:MAX dimers to DNA in vitro (*Figure 6—figure supplement 1A and B*).

To further explore whether the MYC–HCF-1 interaction influences the ability of either protein to engage its chromatin binding sites in cells, we performed ChIP-Seq for HCF-1$_N$ and MYC-HA in our switchable *MYC* cells that were treated with 4-OHT for 24 hr. Binding of HCF-1$_N$ to chromatin is largely unaffected by the 4A or VP16 HBM mutations (*Figure 6A* and *Figure 6—figure supplement 1C*), demonstrating that MYC does not recruit HCF-1 to chromatin. Binding of MYC is subtly altered by both the 4A and VP16 HBM mutations (*Figure 6B* and *Figure 6—figure supplement 1D*), but these changes are widespread and for the most part shared between the loss-of-function and gain-of-function MYC mutants (*Figure 6—source data 1*). Indeed, focusing on the top 140 significant changes, we see that the 4A and the VP16 HBM mutants both tend to have increased or expanded chromatin binding, compared to WT MYC (*Figure 6C*). Visual inspection of the ChIP-Seq data (*Figure 6D* and *Figure 6—figure supplement 1E*) confirms the subtlety of these effects and reinforces the concept that the binding of MYC (and HCF-1) to chromatin is not impacted in opposite ways by the 4A and VP16 HBM mutations. We further verified by ChIP-PCR—at genes flagged as direct targets in *Figure 5*—that binding of MYC (*Figure 6E*) and HCF-1 (*Figure 6—figure supplement 1F*) is largely insensitive to the 4A and VP16 HBM mutations. Thus, although the HBM may play a modest role in chromatin targeting by MYC in cells, this is likely to be independent of the MYC–HCF-1 interaction. We conclude that MYC and HCF-1 interact to control the expression of ribosome biogenesis and mitochondrially connected genes through a co-recruitment-independent mechanism.

## The MYC–HCF-1 interaction is important for tumor engraftment and maintenance

The ability of MYC to regulate ribosome (*van Riggelen et al., 2010*) and mitochondrial (*Morrish and Hockenbery, 2014*) biogenesis are core aspects of its tumorigenic repertoire. We would expect, therefore, that disrupting the MYC–HCF-1 interaction would have a significant impact on the ability of Ramos lymphoma cells to establish and maintain tumors in vivo. To address this expectation, we tested the impact of the 4A MYC mutant on tumorigenesis in mice. Because this is such an aggressive tumor model (*Thomas et al., 2019*), we did not test the gain-of-function VP16

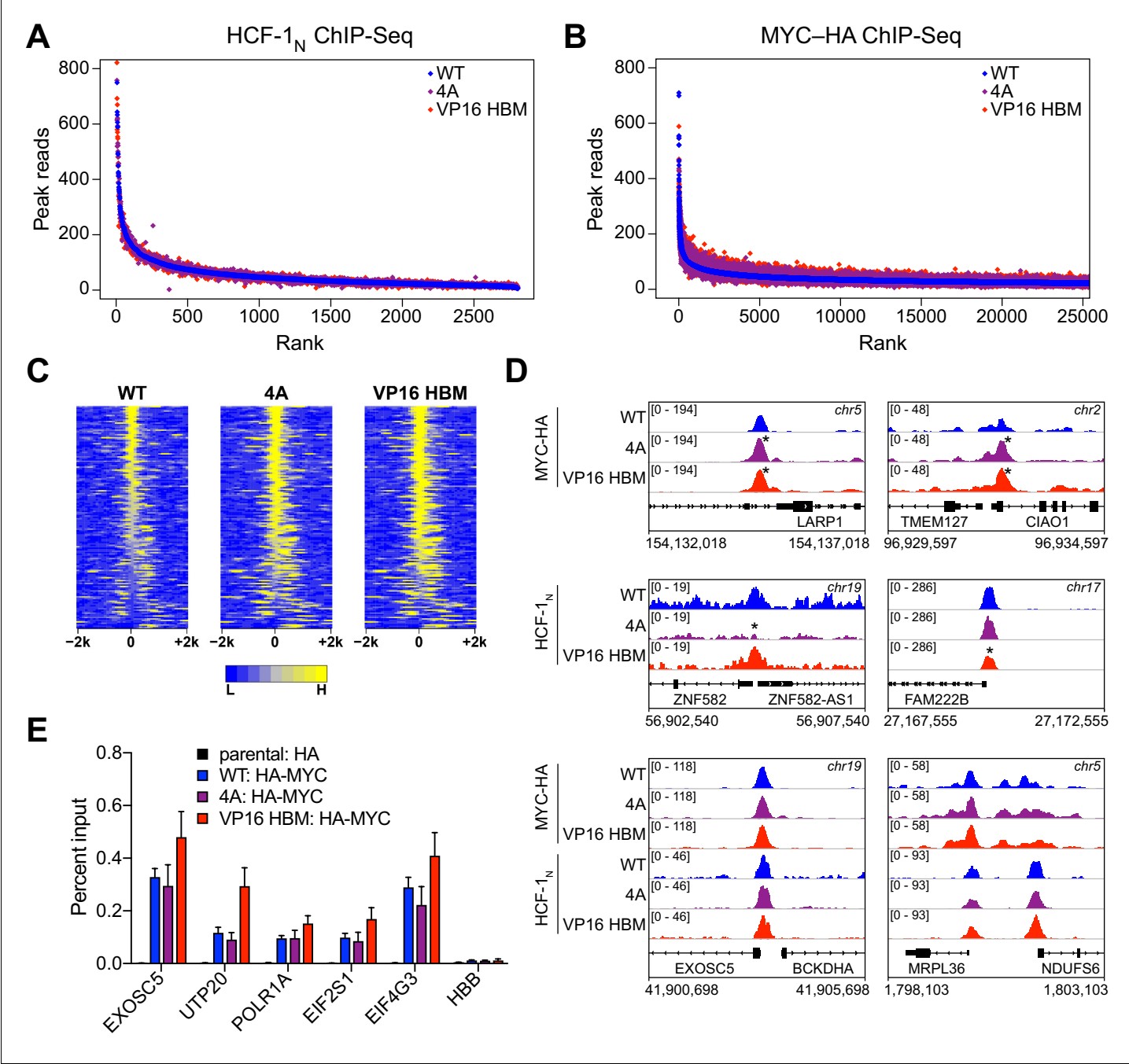

**Figure 6.** MYC and host cell factor (HCF)–1 bind chromatin independent of their ability to interact. (A) Scatterplots of normalized average read counts for HCF-1$_N$ ChIP-seq peaks in wild-type (WT), 4A, or VP16 HCF-1-binding motif (HBM) switched cells. (B) As in (A) but showing normalized average read counts for MYC–HA ChIP-seq peaks. (C) Heatmap of the combined average normalized peak intensity in 100 bp bins for MYC-HA peaks that were significantly changed (false discovery rate [FDR] < 0.05 and |log$_2$FC| > 0.7) for both the 4A and VP16 HBM mutants, and were within ±2 kb of a TSS. (D) Example IGV screenshots of regions that had significant (top) or non-significant (bottom) changes for MYC-HA or HCF-1$_N$ by ChIP-seq. Asterisks mark the peaks that were significantly changed compared to WT. (E) ChIP, using anti-HA antibody, was performed on parental or switchable Ramos cells treated for 24 hr with 20 nM 4-OHT. Enrichment of genomic DNA was monitored by qPCR using primers that amplify across peaks. *HBB* is a negative locus for HA-MYC. ChIP efficiency was measured based on the percent recovery from input DNA. Shown are the mean and standard error for three biological replicates.

The online version of this article includes the following source data and figure supplement(s) for figure 6:

**Source data 1.** MYC-HA ChIP-seq peaks significantly (false discovery rate [FDR] < 0.05) affected by 4A and VP16 host cell factor (HCF)–1-binding motif mutants.

*Figure 6 continued*

**Figure supplement 1.** Impact of host cell factor (HCF)–1-binding motif (HBM) mutations on MYC and HCF-1 binding to chromatin.

HBM mutant. In these experiments, we included a second, independent clone carrying the switchable 4A mutation (4A-1 and 4A-2); we also included a switchable Δ264 mutant (*Thomas et al., 2019*), which deletes residues in the carboxy-terminal half of MYC required for its nuclear localization, as well as interaction with WDR5, HCF-1, and MAX.

First, we assayed tumor engraftment by switching the engineered cells in culture and then injecting into the flanks of nude mice (*Figure 7A*). As expected, the WT to WT switched cells develop tumors rapidly in vivo (*Figure 7B, Figure 7—figure supplement 1A*, and *Figure 7—source data 1*), resulting in all mice reaching humane endpoints and being euthanized by 21 days post-injection (*Figure 7C*). In contrast, 4A-1, 4A-2, and Δ264 switched cells are significantly delayed, both in tumor growth (*Figure 7B, Figure 7—figure supplement 1A*, and *Figure 7—source data 1*) and mortality (*Figure 7C*). Although 4A-1, 4A-2, and Δ264 switched cells did form tumors, these appear to originate from the outgrowth of unswitched cells in the injected populations, as ~75% of cells in these tumors are in their unswitched state (*Figure 7D*). In this assay, therefore, the MYC–HCF-1 interaction is required for tumor growth, and there is little if any difference between disruption of the MYC–HCF-1 interaction and disabling the majority of the nuclear functions of MYC.

Next, we injected unswitched cells into the flanks of mice, allowed tumors to form, and then switched to each of the MYC variants by injecting mice with tamoxifen (*Figure 7E*). As we observed previously (*Thomas et al., 2019*), the 'WT to WT' tumors continue to grow rapidly after switching (*Figure 7F, Figure 7—figure supplement 1B*, and *Figure 7—source data 1*), and all mice had to be euthanized before 30 days (*Figure 7G*). For the 4A switches, however, tumors rapidly regressed (*Figure 7F*), and all mice survived—and were tumor free—for the 60-day duration of the experiment (*Figure 7—figure supplement 1B* and *Figure 7—source data 1*). Regression of the 4A tumors occurs at a pace that is virtually indistinguishable from the Δ264 mutant (*Figure 7—figure supplement 1B* and *Figure 7—source data 1*), and like the Δ264 scenario, is accompanied by high levels of apoptosis, as measured by Annexin V staining (*Figure 7H*), caspase activity (*Figure 7—figure supplement 1C*), and sub-$G_1$ DNA content (*Figure 7—figure supplement 1D*). We conclude that the interaction of MYC with HCF-1 is essential for tumor maintenance in this context.

Finally, we performed RNA-Seq on tumor material excised 48 hr after switching (*Figure 7—figure supplement 1E* and *Figure 7—source data 2*). Thousands of differentially expressed genes were identified in the mutant switch tumors, many of which are shared between the 4A and Δ264 mutants—for both the decreased (*Figure 7—figure supplement 1F*) and induced (*Figure 7—figure supplement 1G*) directions. Common reduced transcripts are enriched in those connected to ribosome biogenesis, translation, and mitochondrial envelope (*Figure 7—figure supplement 1H*), whereas those that are induced are enriched for functions including transcription co-regulator activity, kinase binding, and the vacuole (*Figure 7—figure supplement 1I*). Many of these gene expression changes are likely due to indirect effects of tumor regression. To tease these changes apart from those more closely connected to the MYC–HCF-1 interaction, we overlaid tumor RNA-Seq with that generated for the 4A MYC mutant in vitro (*Figure 3*). Responses that were only observed in the tumor regression model were linked to fairly broad categories such as 'protein binding' or 'mRNA metabolism' (*Figure 7—figure supplement 1J and K*). Responses that were shared between the in vitro and in vivo systems were, in contrast, more extensive and specific. Indeed, more than 70% of the genes repressed in the 4A cell line are also repressed in either 4A-1 or 4A-2 tumors, and there is a common set of 942 genes that are shared between all three data sets (*Figure 7I*). These genes coalesce on those connected to ribosome biogenesis, translation, and the mitochondria (*Figure 7J*). The overlap of induced genes was less pronounced (30%; *Figure 7K*) and these genes are less clustered, although we do observe modest enrichment in categories connected to metabolism, chromatin binding, and transcription coregulator activity (*Figure 7L*). The recurring connections we observe between the MYC–HCF-1 interaction and ribosome biogenesis and mitochondrial function, both in vitro and in vivo, strongly support the notion that a major function of this interaction is to stimulate ribosome production and mitochondrial vigor, and that these actions are central for the ability of MYC to drive tumor onset and maintenance.

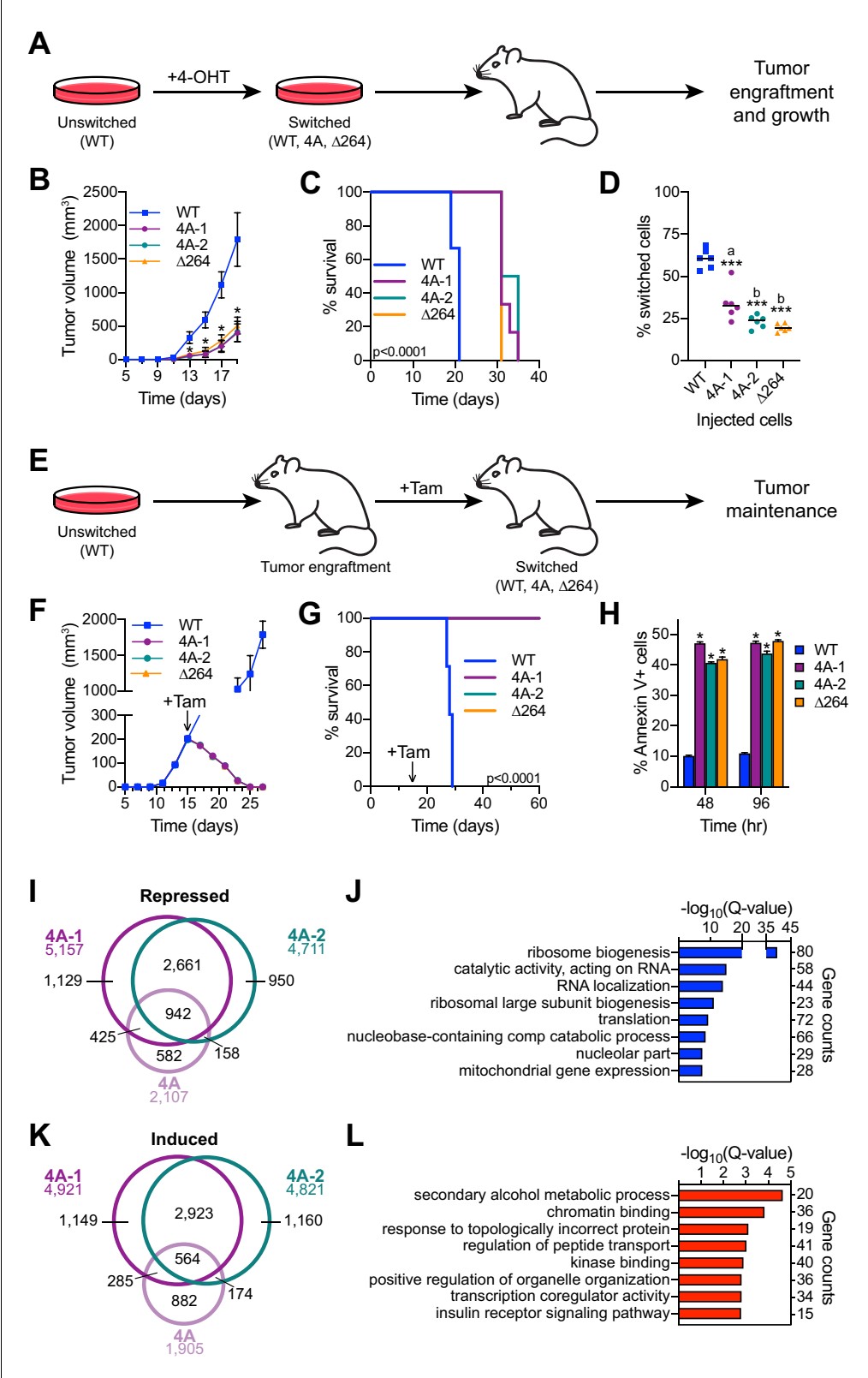

**Figure 7.** The MYC–host cell factor (HCF)–1 interaction is required for tumor engraftment and maintenance. (**A**) Tumor engraftment schema: wild-type (WT), 4A-1, 4A-2, and Δ264 cells are switched in culture prior to injection into flank of nude mice to test the impact of the mutations on tumor engraftment and growth. (**B**) Average tumor volume over time following injection of switched cells. Shown are the mean and standard error for six mice each. Only days 5–19 are shown here; the full course of the experiment is depicted in *Figure 7—figure supplement 1A*. Student's t-test between WT

*Figure 7 continued on next page*

*Figure 7 continued*

and each of the mutants was used to calculate p-value; *p<0.000043. (C) Kaplan–Meier survival curves of mice (n = 6 of each) injected with switched cells. Log-rank test was used to calculate p-value (<0.0001) from six biological replicates. (D) PCR assays of genomic DNA were used to determine the proportion of switched cells present in each tumor after sacrifice. Each dot represents an individual tumor, and the line indicates the mean for each group. Student's t-test between WT and each of the mutants was used to calculate p-values; a = 0.0002, b < 0.0001. (E) Tumor maintenance schema: Unswitched WT, 4A-1, 4A-2, and Δ264 cells were injected into the flanks of nude mice. Tumors were grown until day 15, at which point mice received tamoxifen injections (one per day for 3 days) to induce switching of the cells. (F) Average tumor volume before and after cells were switched. The day at which tamoxifen (Tam) administration was initiated is indicated with an arrow. Shown are the mean and standard error for seven mice for WT and six mice for 4A-1, 4A-2, and Δ264 cells. (G) Kaplan–Meier survival curves of mice in the tumor maintenance assay (n = 7 for WT, and n = 6 for 4A-1, 4A-2, and Δ264). The day at which tamoxifen (Tam) administration was initiated is indicated with an arrow. Log-rank test was used to calculate p-value (<0.0001). (H) Annexin V staining and flow cytometry were performed on cells isolated from tumors at 48 and 96 hr following the first tamoxifen administration to determine the extent of apoptosis. Shown are the mean and standard error for four mice each. Student's t-test between WT and each of the mutants was used to calculate p-value; *p<0.000001. (I) Venn diagram showing the relationship between transcripts significantly (false discovery rate [FDR] < 0.05) decreased in the 4A cell line and the 4A-1 and 4A-2 tumors. (J) Gene ontology (GO) enrichment analysis of transcripts significantly (FDR < 0.05) decreased in the 4A cell line and the 4A-1 and 4A-2 tumors. (K) Venn diagram showing the overlap of transcripts significantly (FDR < 0.05) increased in the 4A cell line and the 4A-1 and 4A-2 tumors. (L) GO enrichment analysis of transcripts significantly (FDR < 0.05) increased in the 4A cell line and the 4A-1 and 4A-2 tumors.

The online version of this article includes the following source data and figure supplement(s) for figure 7:

**Source data 1.** Tumor volumes for engraftment and maintenance assays.
**Source data 2.** Tumor RNA-Seq significant changes.
**Figure supplement 1.** Disruption of the MYC—host cell factor (HCF)–1 interaction in vivo.

# Discussion

The wealth of MYC interaction partners provides a rich resource for the discovery of novel ways to eventually inhibit MYC in the clinic. Unfortunately, the complexity of the MYC interactome also presents a barrier to prioritizing which co-factors to pursue. The highest priority co-factors should be those that play a critical role in the core tumorigenic functions of the protein, and where there is proof-of-concept that disrupting interaction with MYC would provide a therapeutic benefit in the context of an existing malignancy. Here we provide this information for HCF-1. We show that MYC interacts with HCF-1 via a non-canonical HBM, identify roles for the MYC–HCF-1 interaction in the control of genes involved in ribosome biogenesis, translation, and mitochondrial function, and show that loss of the MYC–HCF-1 interaction promotes frank and irreversible tumor regression in vivo. Although we do not yet know if a therapeutic window exists for targeting MYC through HCF-1, and we do not know if our findings will extend to other tumor types, this work, together with our previous study (*Thomas et al., 2016*), highlights HCF-1 as a critical MYC co-factor and one worth pursuing as means to inhibit MYC in cancer.

Through the use of mutations that bidirectionally modulate the interaction between MYC and HCF-1, inducible degradation of HCF-1$_N$, and ChIP-Seq analyses, we identified a relatively small set of genes that we posit are direct targets of the MYC–HCF-1 interaction. It is possible that the multi-pronged strategy we took excludes some bonafide MYC–HCF-1 target genes. But this approach is important because we cannot exclude the possibility that MbIV interacts with multiple factors besides HCF-1. Indeed, we observed instances where the 4A and VP16 HBM mutations produce similar effects on MYC behavior, suggesting that they impact some common aspect of MYC function that is independent of HCF-1. Nonetheless, the genes surviving our stringent criteria can be considered high-confidence targets, and are particularly interesting because of their (i) biological clustering, (ii) connections to core pro-tumorigenic MYC activities, and (iii) ability to account for many of the overt consequences of modulating the MYC–HCF-1 interaction on metabolism and tumorigenesis. What is also interesting about this set of genes is that they appear to be regulated by MYC and HCF-1 through a co-recruitment-independent mechanism.

Among the direct targets of the MYC—HCF-1 interaction are genes that catalyze rate-limiting steps in both ribosome biogenesis (rDNA transcription by POLR1A) and translation (initiator tRNA binding to start codon by EIF2S1 [EIF-2α]) (*Hershey, 1991*; *Laferté et al., 2006*). MYC regulates ribosome biogenesis through controlling and coordinating the transcription of ribosomal DNA, ribosomal protein genes, and components in the processing and assembly of ribosomes (*van Riggelen et al., 2010*). Furthermore, regulation of tRNA ligases by MYC is considered an essential contributor

to MYC-driven cell growth in *Drosophila* (*Zirin et al., 2019*). Interestingly, even with the direct MYC–HCF-targets classified as mitochondrially connected, we see that links to protein synthesis—MARS2, MRPS12, and MRPL32, for example—are specifically involved in the synthesis of mitochondrial proteins, including those required for oxidative phosphorylation. We conclude that, in the context of its relationship with MYC, HCF-1 is dedicated to promoting multiple aspects of biomass accumulation.

The notion that there are process-specific co-factors for MYC is not widely appreciated. MIZ-1 (*Vo et al., 2016*) and WDR5 (*Thomas et al., 2019*) are perhaps the best examples to date; our work here on HCF-1 further reinforces this concept. There are some particularly interesting parallels between the actions of WDR5 and HCF-1 as process-specific co-factors for MYC: both interact with MYC through conserved MYC boxes, both control the expression of relatively small cohorts of genes, both are required to interact with MYC to establish and maintain tumors, and both have clear links to biomass accumulation. Beyond this point, however, these parallels break down. There is little overlap of target genes regulated by the MYC–WDR5 and MYC–HCF-1 interactions, and whereas MYC and HCF-1 associate with predominantly control ribosome biogenesis, MYC and WDR5 work together to stimulate the expression of genes encoding structural components of the ribosome (*Thomas et al., 2019*). This division of labor between HCF-1 and WDR5 in different aspects of protein synthesis is particularly intriguing given that HCF-1 and WDR5 work together as part of multi-protein chromatin modifying complexes (*Tyagi et al., 2007*; *Wysocka et al., 2003*). Yet MYC interacts with each protein through separate MYC boxes, and indeed the way in which MYC interacts with WDR5 likely precludes WDR5 from assembling into canonical HCF-1-containing complexes (*Thomas et al., 2015*). We believe the separation of interaction surfaces allows MYC to access non-canonical functions of both WDR5 and HCF-1, and may have evolved to permit discrete regulation of the constituents of the ribosomes versus the factors required to assemble these constituents into ribosome particles. Indeed, because HCF-1 (*Mazars et al., 2010*), MYC (*Chou et al., 1995*), and the MYC–HCF-1 interaction (*Itkonen et al., 2019*) all have ties to the metabolic sensor OGT (*Swamy et al., 2016*), it is possible that this separation of function allows for rapid modulation of ribosome assembly by HCF-1 when the metabolic state of the cell declines, while at the same time ensuring that ribosome subunits are present for rapid recovery when metabolism ramps up.

The concept that HCF-1 is a biomass-specific co-factor for MYC can also account for our discovery that intracellular amino acid levels are increased when the MYC–HCF-1 interaction is disrupted. Impairment of ribosome biogenesis and translation can lead to an accumulation of amino acids and compensatory changes in the expression of their transporters (*Guan et al., 2014*; *Scott et al., 2014*). Of note, the way in which amino acid levels almost globally respond to changes in the MYC–HCF-1 interaction can also provide a simple explanation for how the MYC–HCF-1 interaction contributes to the glutamine addiction of Ramos cells. If MYC partners with HCF-1 to drive biomass accumulation, the net effect of this interaction will be to increase the demand for intracellular amino acids, with glutamine playing a particularly important role in the biosynthesis of multiple non-essential amino acids, including glutamate, arginine, proline, and alanine (*Hosios et al., 2016*). The ability of MYC to drive glutamine-addiction is a defining characteristic of its tumorigenic repertoire (*Tansey, 2014*), and is thought to result from the ability of MYC to promote glutaminolysis and induce the expression of amino acid transporters (*Wise et al., 2008*). But the accumulation of glutamine we observe upon disruption of the MYC–HCF-1 interaction, together with the specific role HCF-1 plays in MYC function, suggests that glutamine addiction might also be fueled by the well-established ability of MYC to stimulate protein synthesis (*Iritani and Eisenman, 1999*).

Our discovery that MYC and HCF-1 work together to regulate transcription via a co-recruitment-independent mechanism is surprising, but not without precedent. E2F transcription factors interact with HCF-1 through a canonical HBM to control the expression of genes connected to cell cycle progression, yet depletion of these proteins has little effect on the recruitment of either to chromatin (*Iwata et al., 2013*; *Parker et al., 2014*; *Tyagi et al., 2007*). Similarly, *Drosophila* Myc and HCF interact to activate transcription and control growth, but their respective interactions with chromatin are independent of one another (*Furrer et al., 2010*). A co-recruitment-independent mechanism may thus be a common way in which transcription factors interact with HCF-1 to modulate transcription. In imagining how this might occur, it is interesting to note that the biochemical context in which HCF-1 exists can be influenced by client proteins; its interaction with E2F1 drives association with SETD1A complexes, whereas its interaction with E2F4 favors binding to the SIN3A HDAC complex (*Tyagi et al., 2007*). Perhaps, therefore, the post-recruitment interaction of MYC and HCF-1

modulates transcription by promoting the ejection of inhibitory proteins like SIN3A from, or recruiting activating proteins like SETD1A to, HCF-1. Further experimentation will be required to determine the mechanism of action.

Finally, there are two additional cancer-relevant connections worth mentioning. First, the COSMIC database (*Tate et al., 2019*) identifies three cancer-associated mutations within the MYC HBM, all of which convert the subprime 'QHNY' motif to 'EHNY', which matches the perfect HBM consensus and closely resembles the gain-of-function VP16 HBM we used in this study. It is possible, therefore, that these rare mutations in MYC could contribute to disease progression via enhancement of the MYC–HCF-1 interaction. This is an intriguing idea, and one worth testing in a less aggressive in vivo model where increases in the tumorigenic potential of MYC can be visualized. Second, our demonstration that disrupting the MYC–HCF-1 interaction in the context of an existing tumor promotes its regression provides compelling proof-of-concept for the idea that inhibitors of this interaction could have utility as anticancer agents. Switching WT MYC to the 4A mutant caused rapid and widespread induction of apoptosis and was associated with changes in the expression of genes connected to ribosome biogenesis, translation, and the mitochondria, consistent with the idea that reduced expression of these MYC–HCF-1 target genes triggers the regression process (*Hanahan and Weinberg, 2011*; *Pelletier et al., 2018*). The small and well-defined nature of the HBM suggests that if structural information becomes available for the HCF-1 VIC domain, it could be possible to develop small molecule inhibitors that block the MYC–HCF-1 interaction. The most obvious concern with this strategy is that HCF-1 is not a MYC-specific co-factor, and that its interactions with other transcription factors may limit or prevent attainment of a therapeutic window. To our knowledge, MYC proteins and E2F3a are the only transcription factors that interact with HCF-1 via an 'imperfect' HBM, which we show here is sub-optimal for robust HCF-1 association. It might be possible to develop a therapeutic window by exploiting the non-canonical nature of the HBM in MYC, with the expectation that this interaction will be more sensitive than others that carry higher affinity HBM motifs. We note, however, that many of the factors with which HCF-1 interacts via an HBM are inherently pro-proliferative, with the E2F proteins in particular playing a predominant role in cancer initiation, maintenance, and response to therapies (*Kent and Leone, 2019*). We also note that HCF-1 has been reported to be overexpressed in cancer, and its overexpression can correlate with poor clinical outcomes (*Glinsky et al., 2005*). It is possible, therefore, that on-target collateral consequences of inhibiting the MYC–HCF-1 interaction could also have therapeutic benefit against malignancies.

# Materials and methods

## Key resources table

| Reagent type (species) or resource | Designation | Source or reference | Identifiers | Additional information |
|---|---|---|---|---|
| Gene (*Homo sapiens*) | *MYC* | NA | UniProt ID: P01106 | |
| Gene (*Homo sapiens*) | *HCFC1* | NA | UniProt ID: P51610 | |
| Strain, strain background (*Escherichia coli*) | Rosetta (DE3) Competent Cells | Millipore | Cat#: 70954 | |
| Strain, strain background (*Escherichia coli*) | XL1-Blue Competent Cells | Agilent Technologies | Cat#: 200249 | |
| Cell line (*Homo sapiens*) | Ramos | ATCC | Cat#: CRL-1596; RRID:CVCL_0597 | |
| Cell line (*Homo sapiens*) | Ramos Cre-ERT2 | *Thomas et al., 2019* | NA | |
| Cell line (*Homo sapiens*) | Ramos Cre-ERT2 MYC-WT | *Thomas et al., 2019* | NA | |

*Continued on next page*

*Continued*

| Reagent type (species) or resource | Designation | Source or reference | Identifiers | Additional information |
|---|---|---|---|---|
| Cell line (*Homo sapiens*) | Ramos Cre-ERT2 MYC-Δ264 | *Thomas et al., 2019* | NA | |
| Cell line (*Homo sapiens*) | Ramos Cre-ERT2 MYC-4A | This paper | NA | See section 'Generation of switchable MYC allele Ramos cell lines' |
| Cell line (*Homo sapiens*) | Ramos Cre-ERT2 MYC-VP16 HBM | This paper | NA | See section 'Generation of switchable MYC allele Ramos cell lines' |
| Cell line (*Homo sapiens*) | Ramos FKBP$^{FV}$-HCF-1$_N$ | This paper | NA | See section 'Generation of dTAG Ramos cell lines' |
| Antibody | Rabbit anti-HCF-1$_C$ polyclonal | Bethyl Laboratories | Cat#: A301-399A; RRID:AB_961012 | Western blotting (1:1000) |
| Antibody | Rabbit anti-HCF-1$_N$ polyclonal | *Machida et al., 2009* | NA | Western blotting (1:10,000), ChIP (6 µl) |
| Antibody | Rabbit anti-c-MYC (Y69) monoclonal | Abcam | Cat#: ab32072; RRID:AB_731658 | Western blotting (1:10,000), IP (2 µg) |
| Antibody | Rabbit anti-HA (C29F4) monoclonal | Cell Signaling Technology | Cat#: 3724; RRID:AB_1549585 | Western blotting (1:4000), ChIP (5 µl), IF (1:500) |
| Antibody | Rabbit anti-WDR5 (D9E1I) monoclonal | Cell Signaling Technology | Cat#: 13105; RRID:AB_2620133 | Western blotting (1:1000) |
| Antibody | Rabbit anti-IgG polyclonal | Cell Signaling Technology | Cat#: 2729; RRID:AB_1031062 | IP (2 µg), ChIP (0.8 µg) |
| Antibody | Mouse anti-FLAG (M2) monoclonal (HRP) | Sigma-Aldrich | Cat#: A8592; RRID:AB_439702 | Western blotting (1:10,000 or 1:50,000) |
| Antibody | Mouse anti-T7 monoclonal (HRP) | Millipore | Cat#: 69048; RRID:AB_11212778 | Western blotting (1:10,000) |
| Antibody | Mouse anti-GAPDH (GA1R) monoclonal (HRP) | Thermo Fisher Scientific | Cat#: MA5-15738-HRP; RRID:AB_2537659 | Western blotting (1:50,000) |
| Antibody | Rabbit anti-GAPDH (D16H11) monoclonal (HRP) | Cell Signaling Technology | Cat#: 8884; RRID:AB_11129865 | Western blotting (1:2500) |
| Antibody | Rabbit anti-α-Tubulin (11H10) monoclonal (HRP) | Cell Signaling Technology | Cat#: 11H10; RRID:AB_10695471 | Western blotting (1:5000) |
| Antibody | Rabbit anti-histone H3 (D1H2) XP monoclonal (HRP) | Cell Signaling Technology | Cat#: 12648; RRID:AB_2797978 | Western blotting (1:5000) |
| Antibody | Goat anti-rabbit IgG polyclonal (HRP) | Thermo Fisher Scientific | Cat#: 31463; RRID:AB_228333 | Western blotting (1:5000) |
| Antibody | Mouse anti-rabbit IgG monoclonal, light chain specific (HRP) | Jackson ImmunoResearch Labs | Cat#: 211-032-171; RRID:AB_2339149 | Western blotting (1:5000) |
| Antibody | Goat anti-rabbit IgG polyclonal (Alexa Fluor 594) | Thermo Fisher Scientific | Cat#: A11012; RRID:AB_2534079 | IF (1:350) |
| Other | Anti-FLAG M2 affinity gel | Sigma-Aldrich | Cat#: A2220; RRID:AB_10063035 | IP (20 µl bed volume) |

*Continued on next page*

*Continued*

| Reagent type (species) or resource | Designation | Source or reference | Identifiers | Additional information |
|---|---|---|---|---|
| Other | Protein A Agarose | Roche | Cat#: 11134515001 | IP/ChIP (20 µl bed volume) |
| Recombinant DNA reagent | pCRIS-mCherry-FLAG-dTAG-HCFC1 | This paper | NA | See section 'Generation of dTAG Ramos cell lines' |
| Recombinant DNA reagent | pGuide-HCFC1-N | This paper | NA | See section 'Generation of dTAG Ramos cell lines' |
| Recombinant DNA reagent | pX330-U6-Chimeric_BB-CBh-hSpCas9 | Addgene | Cat#: 42230; RRID:Addgene_42230 | |
| Recombinant DNA reagent | MYC-WT targeting vector | *Thomas et al., 2019* | NA | |
| Recombinant DNA reagent | MYC-4A targeting vector | This paper | NA | See section 'Generation of switchable MYC allele Ramos cell lines' |
| Recombinant DNA reagent | MYC-VP16 HBM targeting vector | This paper | NA | See section 'Generation of switchable MYC allele Ramos cell lines' |
| Recombinant DNA reagent | pGuide-MYC1 | This paper | NA | See section 'Generation of switchable MYC allele Ramos cell lines' |
| Recombinant DNA reagent | pFLAG-C2 | *Thomas et al., 2015* | NA | |
| Recombinant DNA reagent | pFLAG-MYC WT | *Thomas et al., 2015* | NA | |
| Recombinant DNA reagent | pFLAG-MYC 4A | This paper | NA | See section 'Transient transfection, western blotting, and immunoprecipitation' |
| Recombinant DNA reagent | pFLAG-MYC H307G | This paper | NA | See section 'Transient transfection, western blotting, and immunoprecipitation' |
| Recombinant DNA reagent | pFLAG-MYC VP16 HBM | This paper | NA | See section 'Transient transfection, western blotting, and immunoprecipitation' |
| Recombinant DNA reagent | pFLAG-MYC VP16 HBM H307G | This paper | NA | See section 'Transient transfection, western blotting, and immunoprecipitation' |
| Recombinant DNA reagent | pFLAG-MYC WBM | *Thomas et al., 2015* | NA | |
| Recombinant DNA reagent | pSUMO-MYC WT-FLAG | *Thomas et al., 2015* | NA | |
| Recombinant DNA reagent | pSUMO-MYC 4A-FLAG | This paper | NA | See section 'In vitro binding assays' |
| Recombinant DNA reagent | pSUMO-MYC VP16 HBM-FLAG | This paper | NA | See section 'In vitro binding assays' |

*Continued on next page*

*Continued*

| Reagent type (species) or resource | Designation | Source or reference | Identifiers | Additional information |
|---|---|---|---|---|
| Recombinant DNA reagent | pT7-IRES His-T7-HCF-1VIC | This paper | NA | See section 'In vitro binding assays' |
| Recombinant DNA reagent | pRSET-6XHis-MYC WT | *Farina et al., 2004* | NA | |
| Recombinant DNA reagent | pRSET-6XHis-MYC 4A | This paper | NA | See section 'Electrophoretic mobility shift assays' |
| Recombinant DNA reagent | pRSET-6XHis-MYC VP16 HBM | This paper | NA | See section 'Electrophoretic mobility shift assays' |
| Commercial assay or kit | Q5 DNA Polymerase | NEB | Cat#: M0491 | |
| Commercial assay or kit | OneTaq DNA Polymerase | NEB | Cat#: M0480 | |
| Commercial assay or kit | Gibson Assembly Cloning Kit | NEB | Cat#: E5510 | |
| Commercial assay or kit | LightShift Chemiluminescent EMSA Kit | Thermo Fisher | Cat#: 20148 | |
| Chemical compound, drug | DMSO | Sigma-Aldrich | Cat#: D2650 | |
| Chemical compound, drug | dTAG-47 | Vanderbilt Institute of Chemical Biology Synthesis Core | | |
| Software, algorithm | PRISM 8 | GraphPad | RRID:SCR_002798 | https://www.graphpad.com/scientific-software/prism/ |
| Software, algorithm | FACSDiva 8.0 | BD Biosciences | RRID:SCR_001456 | https://www.bdbiosciences.com/en-us/instruments/research-instruments/research-software/flow-cytometry-acquisition/facsdiva-software |
| Software, algorithm | FlowJo | FlowJo | RRID:SCR_008520 | https://www.flowjo.com/ |
| Software, algorithm | MetaboAnalyst 4.0 | *Chong et al., 2019* | RRID:SCR_015539 | https://www.metaboanalyst.ca/ |
| Software, algorithm | Metascape | *Zhou et al., 2019* | RRID:SCR_016620 | https://metascape.org/ |
| Software, algorithm | DAVID | *Huang et al., 2009a, Huang et al., 2009b* | RRID:SCR_001881 | https://david.ncifcrf.gov/ |

## Primers and cloning

See *Supplementary file 1* for primer sequences. PCRs were performed using either Q5 DNA Polymerase (NEB, Ipswich, Massachusetts, M0491) or OneTaq DNA Polymerase (NEB M0480). Gibson assemblies were performed using Gibson Assembly Cloning Kit (NEB E5510). More specific details about cloning steps can be found in the relevant sections.

## Cell culture

293T cells were maintained in DMEM with 4.5 g/l glucose, L-Glutamine, and sodium pyruvate (Corning, Corning, New York, 10–013-CV), and supplemented with 10% fetal bovine serum (FBS, Denville Scientific, Metuchen, New Jersey, FB5001-H), and 1% penicillin/streptomycin (P/S, Gibco, Waltham, Massachusetts, 15140122). Ramos cells were obtained directly from the ATCC (Manassas, Virginia, CRL-1596) and maintained in RPMI 1640 with L-glutamine (Corning 10–040-CV), and supplemented with 10% FBS (Denville Scientific FB5001-H), and 1% P/S (Gibco 15140122). All cell lines used were confirmed as mycoplasma-negative; 293T cells were authenticated by STR profiling.

## Antibodies

Rabbit anti-HCF1$_C$ polyclonal (Bethyl Laboratories, Montgomery, Texas, Cat# A301-399A); rabbit anti-HCF1$_N$ polyclonal (*Machida et al., 2009*); rabbit anti-c-MYC (Y69) monoclonal (Abcam, Cambridge, United Kingdom, Cat# ab32072); rabbit anti-HA (C29F4) monoclonal (Cell Signaling Technology, Danvers, Massachusetts, Cat# 3724); rabbit anti-WDR5 (D9E1I) monoclonal (Cell Signaling Technology Cat# 13105); rabbit anti-IgG polyclonal (Cell Signaling Technology Cat# 2729); mouse anti-FLAG (M2) monoclonal, HRP-conjugated (Sigma-Aldrich, St. Louis, Missouri, Cat# A8592); mouse anti-T7 monoclonal, HRP conjugated (Millipore, Burlington, Massachusetts, Cat# 69048); mouse anti-GAPDH (GA1R) monoclonal, HRP conjugated (Thermo Fisher Scientific, Waltham, Massachusetts, Cat# MA5-15738-HRP); rabbit anti-GAPDH (D16H11) monoclonal, HRP conjugated (Cell Signaling Technology Cat# 8884); goat anti-rabbit IgG polyclonal, HRP conjugated (Thermo Fisher Scientific Cat# 31463); mouse anti-rabbit IgG monoclonal, light chain specific, HRP conjugated (Jackson ImmunoResearch Laboratories, West Grove, Pennsylvania, Cat# 211-032-171); rabbit anti-histone H3 (D1H2) XP monoclonal, HRP conjugated (Cell Signaling Technology Cat# 12648); rabbit anti-α-Tubulin (11H10) monoclonal, HRP conjugated (Cell Signaling Technology Cat# 9099); goat anti-rabbit IgG polyclonal, Alexa Fluor 594 conjugated (Thermo Fisher Scientific A11012).

## Generation of switchable MYC allele Ramos cell lines

Q5 site-directed mutagenesis of MYC-WT targeting vector from *Thomas et al., 2019* was used to create MYC-4A (4A_F and 4A_R) and MYC-VP16 HBM (VP16 HBM_F and VP16 HBM_R). The pGuide plasmid described by *Thomas et al., 2019* was used as a backbone to introduce the sgRNA sequence GCTACGGAACTCTTGTGCGTA (pGuide-MYC1) by Q5 site directed mutagenesis with the primers GUIDE MYC-1A and GUIDE MYC-1B.

For the generation of switchable cells, 10 million Ramos cells stably expressing CRE-ER$^{T2}$ (*Thomas et al., 2019*) were electroporated (BioRad Gene Pulser II, 220 V and 950 µF) with 10 µg of relevant targeting vector (MYC-4A or MYC-VP16 HBM), 15 µg pGuide-MYC1, and 15 µg pX330-U6-Chimeric_BB-CBh-hSpCas9 (gift from Feng Zhang, AddGene plasmid #42230) (*Cong et al., 2013*). WT and Δ264 cell lines were the same as those used in *Thomas et al., 2019*. Cells were treated with 150 ng/ml puromycin (Sigma-Aldrich P7255) and 100 µg/ml hygromycin (Corning 30240CR), selecting for the switchable MYC cassette and CRE-ER$^{T2}$ recombinase, respectively. Following selection, single cells, stained using propidium iodide (PI, Sigma-Aldrich P4864) for viability, were sorted by the Vanderbilt Flow Cytometry Shared Resource using a BD FACSAria III flow cytometer into a 96-well plate to generate clonal cell lines under puromycin and hygromycin selection. Individual clones were expanded, and initially validated by switching for 24 hr using 20 nM (Z)−4-Hydroxytamoxifen (4-OHT, Tocris, Minneapolis, Minnesota, 3412), and flow cytometry (see below) for GFP expression. Further validation was performed by western blotting after 24 hr 20 nM ±4 OHT (see below) and by sSouthern blotting (see below). The 4A cell line (4A-1) used for the majority of experiments is haploinsufficient for part of chromosome 11, approximately between co-ordinates 118,685,194 and 134,982,408. This region of the genome was excluded from genomic analyses. For all experiments, switching was performed by treatment with 20 nM 4-OHT for 2 or 24 hr (see relevant method or figure legend).

## Generation of dTAG Ramos cell lines

To create pCRIS-mCherry-FLAG-dTAG-HCFC1, pCRIS-PITChv2-Puro-dTAG (BRD4) (gift from James Bradner, AddGene plasmid #91793) (*Nabet et al., 2018*) was first modified to remove the BRD4 homology arms and replace the 2XHA tags with a FLAG tag. This was done by Gibson assembly of

the vector (Q5 amplification using pCRIS-HCFC1N_F and pCRIS_R), puromycin cassette (Q5 amplification using Puro-1 and Puro-FLAG_R), and FKBP12$^{FV}$ (Q5 amplification using FLAG-FKBP_F and pCRIS-FKBP_R). The resulting vector was again modified using Gibson assembly by combining the vector (Q5 amplification using pCRIS-HCFC1N_F and pCRIS-HCFC1N_R), an upstream 271 bp *HCFC1* 5' homology arm (hg19 chrX:153236265–153236535, OneTaq amplification using HCFC1N_F and HCFC1N-5'Hom_R), mCherry (Q5 amplification using mCherry_F and mCherry_R), FKBP12$^{FV}$ (Q5 amplification using HCFC1-mCherry-FKBP_F and HCFC1-mCherry-FKBP_R), and a downstream 800 bp *HCFC1* 3' homology arm (hg19 chrX:153235465–153236264, OneTaq amplification using HCFC1N-3'Hom_F and HCFC1N-3'Hom_R). The pGuide plasmid described by *Thomas et al., 2019* was used as a backbone to introduce the guide RNA sequence CAGAAGCACCGCTGGCAAGT (pGuide-HCFC1-N) by Q5 site-directed mutagenesis with the primers HCFC1N-sgRNA_F and HCFC1N-sgRNA_R.

Fifteen micrograms of pGuide-HCFC1-N and 10 µg of pCRIS-mCherry-FLAG-dTAG-HCFC1 were electroporated (BioRad, Hercules, California, Gene Pulser II, 220 V and 950 µF) into Ramos cells, alongside 15 µg pX330-U6-Chimeric_BB-CBh-hSpCas9 (gift from Feng Zhang, AddGene plasmid #42230) (*Cong et al., 2013*) into 10 million Ramos cells. Because *HCFC1* is on the X chromosome and Ramos cells are XY (*Klein et al., 1975*), only a single copy of *HCFC1* is present for targeting using CRISPR/Cas9. Following electroporation, cells were expanded and a population of mCherry-positive cells, stained using Zombie NIR viability dye (BioLegend, San Diego, California, 423105), was sorted by the Vanderbilt Flow Cytometry Shared Resource using a FACSAria III flow cytometer (Becton Dickinson (BD), Franklin Lakes, New Jersey). This population of cells was expanded further before validation by western blotting. All experiments were conducted using this population and were treated with either DMSO (Sigma-Aldrich D2650) or 500 nM dTAG-47, which was synthesized by the Vanderbilt Institute of Chemical Biology Synthesis Core.

## Transient transfection, western blotting, and immunoprecipitation

Q5 site-directed mutagenesis of pFLAG-MYC WT (*Thomas et al., 2015*) was used to generate the following plasmids: pFLAG-MYC 4A (4A_F and 4A_R), pFLAG-MYC H307G (H307G_F and H307G_R), pFLAG-MYC VP16 HBM (VP16 HBM_F and VP16 HBM_R), and pFLAG-MYC VP16 HBM H307G (VP16 HBM H307G_F and VP16 HBM H307G_R). pFLAG-MYC WBM is from *Thomas et al., 2015*. Plasmid (19 µg) was prepared with 0.25 M CaCl$_2$, incubated for 10 min with 1× HBS (2× HBS: 140 mM NaCl, 1.5 mM Na$_2$HPO$_4$, 50 mM HEPES, pH 7.05), and then applied drop-wise to 293T cells. Cells were grown for 2 days before harvesting (see below).

Cell lysates for western blotting or immunoprecipitation (IP) were prepared by rinsing cells twice in ice-cold 1× PBS, and harvesting in Kischkel Buffer (50 mM Tris pH 8.0, 150 mM NaCl, 5 mM EDTA, 1% Triton X-100) + protease inhibitor cocktail (PIC, Roche, Basel, Switzerland, 05056489001). Cells were sonicated at 25% power for 10 s (Cole-Parmer, Vernon Hills, Illinois, GE 130PB-1), debris removed by centrifugation, and protein concentration determined using Protein Assay Dye (BioRad 500–0006) against a bovine serum albumin (BSA) standard. For western blotting, lysate was diluted in 5× Laemmli Buffer (375 mM Tris pH 6.8, 40% glycerol, 10% SDS, bromophenol blue, 2-Mercaptoethanol). For IP, the concentration of samples was balanced using Kischkel Buffer. Antibody or anti-FLAG M2 Affinity Gel (Millipore A2220) was added, and samples rotated overnight at 4℃. For unconjugated antibodies, 20 µl bed volume of Protein A Agarose (Roche 11134515001) was added the following day to each sample, and rotated for 2–4 hr at 4℃. Samples were then washed 4× with Kischkel buffer (2× 4℃ and 2× at room temperature), and incubated in Laemmli buffer for 5 min at 95℃.

Protein from lysates and IPs were separated out by SDS-polyacrylamide gel electrophoresis (PAGE) in running buffer (25 mM Tris, 192 mM glycine, 0.2% SDS). Wet transfer to PVDF (PerkinElmer, Waltham, Massachusetts, NEF1002) was carried out in Towbin Transfer Buffer (25 mM Tris, 192 mM glycine, and 10% methanol). Membrane was blocked in 5% milk in TBS-T (20 mM Tris pH 7.6, 140 mM NaCl, 0.1% Tween-20), hybridized overnight in primary antibody (or 1 hr for HRP-conjugated), and for 1 hr in HRP-conjugated secondary antibody (if required). ECL substrates, SuperSignal West Pico (Pierce, Waltham, Massachusetts, 34080), Pico+ (Pierce 34580), and Femto (Pierce 34095) were used in various combinations for detection of bands by exposure to film.

## Chromatin fractionation

Chromatin fractionation was performed, with slight modification, as described by *Méndez and Stillman, 2000*. Switchable MYC Ramos cells that had been treated for 24 hr with 20 nM 4-OHT were washed in ice-cold PBS, resuspended in 200 μl Buffer A (10 mM HEPES pH 7.9, 10 mM KCl, 1.5 mM MgCl$_2$, 0.34 M sucrose, 10% glycerol, 1 mM DTT) + PIC + PMSF + 0.1% Triton X-100, and incubated on ice for 10 min. The resulting lysate was centrifuged 1300 × g for 5 min at 4°C, with the pellet (P1) containing the nuclei. The supernatant (S1) was centrifuged at 20,000 × g for 10 min at 4°C, giving pellet P2 and supernatant S2. P2 was discarded and S2 corresponding to the soluble portion of the total cell extract was diluted out in Laemmli buffer. P1 was gently washed in Buffer A, resuspended by pipetting up and down in Buffer B (3 mM EDTA, 0.2 mM EGTA, 1 mM DTT) + PIC + PMSF, and incubated on ice for 30 min. The lysed nuclei were centrifuged at 1700 × g for 5 min at 4°C to give soluble nuclear proteins (S3) and chromatin-bound proteins (P3). S3 was diluted out in Laemmli buffer. P3 was gently washed in Buffer B, resuspended in Laemmli buffer, and sonicated for 15 s at 25% power. All samples were incubated at 95°C for 3 min. Proteins were separated out by SDS-PAGE and transferred to PVDF, as described above, and probed for HCF-1$_C$, HA, tubulin, and H3.

## Immunofluorescence

Following a 24 hr treatment with 20 nM 4-OHT, 10$^5$ switchable MYC Ramos cells were attached to slides by CytoSpin (800 RPM, 3 min), fixed for 10 min with 3% methanol-free formaldehyde (Thermo Fisher Scientific 28908) diluted in PBS, and washed three times with PBS. Cells were permeabilized for 10 min with Permeabilization Solution (0.1% Triton X-100 in PBS), blocked for 1 hr with Blocking Solution (2.5% BSA in Permeabilization Solution), and incubated with Blocking Solution containing anti-HA antibody (1:500, Cell Signaling 3724) for 1 hr. Cells were washed three times with PBS and incubated with Blocking Solution containing Goat anti-Rabbit IgG Alexa Fluor 594 antibody (1:350, Thermo Fisher Scientific A11012) for 1 hr. Cells were then washed three times with PBS, and coverslips mounted with ProLong Gold Antifade Mountant with DAPI (Thermo Fisher Scientific P36941). Slides were imaged by wide-field fluorescent microscopy on a Nikon Eclipse Ti equipped with a Nikon Plan Apo λ 100×/1.45 Oil objective, Nikon DS-Qi2 camera, and Excelitas X-Cite 120LED illuminator using identical settings for each sample and representative images shown.

## Flow cytometry and cell cycle analysis

Cells were filtered into 35 μm nylon mesh Falcon round bottom test tubes for flow cytometry, which was performed in the Vanderbilt Flow Cytometry Shared Resource. Single cells were gated based on side and forward scatter using the stated instruments.

To determine the proportion of switching of the switchable *MYC* allele Ramos cell lines, cells were fixed in 1% formaldehyde (FA, Thermo Fisher 28908) in PBS for 10 min at room temperature. The number of GFP-positive cells was determined using a BD LSR II flow cytometer.

For cell cycle analysis of the switchable MYC allele Ramos cell lines, cells were treated with 4-OHT for 2 hr, at which point the media was replaced. Cells were maintained for 7 days, at which point 1 × 10$^6$ cells were collected and fixed in 1% FA (Thermo Fisher 28908) in PBS for 10 min at room temperature, then washed 2× with PBS. Permeabilization and staining was done using cell cycle staining buffer (PBS, 10 μg/ml PI [Sigma-Aldrich P4864], 100 μg/ml RNAse A, 2 mM MgCl$_2$, 0.1% Triton X-100) for 25 min at room temperature, then stored overnight at 4°C. PI staining of at least 10,000 single cells for each of the GFP-negative (GFP−) and GFP-positive (GFP+) populations was measured using a BD LSR Fortessa, and cell cycle distribution determined using BD FACSDIVA software.

For cell cycle analysis of the FKBP$^{FV}$-HCF-1$_N$ Ramos cells, cells were treated with DMSO or 500 nM dTAG for 24 hr. One million cells were collected and fixed in ethanol overnight. After fixation, cells were stained overnight at 4°C in cell cycle staining buffer (PBS, 10 μg/ml PI [Sigma-Aldrich P4864], 100 μg/ml RNAse A, 2 mM MgCl$_2$). PI staining of at least 10,000 single cells was measured using a BD LSR Fortessa, and cell cycle distribution determined using BD FACSDIVA software.

See section 'In vivo studies: Tumor formation and maintenance assays' for details regarding flow cytometry experiments conducted on cells extracted from mice tumors.

## Southern blotting

Genomic DNA (gDNA) was prepared from parental and unswitched MYC switchable cells (WT, 4A, and VP16 HBM) (Miller et al., 1988). Briefly, cells were rinsed in ice-cold 1× PBS and resuspended in DNA extraction buffer (10 mM Tris pH 8.1, 400 mM NaCl, 10 mM EDTA, 1% SDS, 50 μg/ml proteinase K [PK, Macherey-Nagel, Düren, Germany, 740506]). Lysis was performed overnight in a rotisserie at 56°C, before gDNA was extracted using ethanol precipitation. Southern blot was performed similar to that described by Southern, 1975. gDNA (10 μg) was digested using XbaI (NEB R0145) and run out on a 1% agarose gel. DNA was transferred overnight to Hybond-N+ nylon membrane (GE Healthcare, Chicago, Illinois, RPN303B) by capillary action in transfer buffer (0.5 M NaOH, 0.6 M NaCl). The following day the membrane was immersed in neutralization buffer (1 M NaCl, 0.5 M Tris pH 7.4), UV cross-linked, and pre-hybridized overnight at 42°C in pre-hybridization buffer (50% formamide, 5× SSCPE [20× SSCPE: 2.4 M NaCl, 0.3 M Na citrate, 0.2 M $KH_2PO_4$, 0.02 M EDTA], 5× Denhardt's solution (Invitrogen, Waltham, Massachusetts, 750018), 0.5 mg/ml salmon sperm DNA [Agilent, Santa Clara, California, 201190], 1% SDS). Templates for probe generation were prepared by Q5 amplification from MYC-WT targeting vector using primers GFP_F and GFP_R (GFP template) and from parental Ramos cell gDNA using primers 5'_F and 5'_R, followed by gel purification. Probes were prepared by random priming of corresponding PCR products (5' and GFP templates) in the presence of [αP32]CTP (PerkinElmer BLU513H100UC). Unincorporated nucleotides were removed using a Sephadex G-50 column (GE Healthcare 28-9034-08). The membrane was incubated overnight at 42°C with probe in hybridization buffer (50% formamide, 5× SSCPE, 5× Denhardt's solution [Invitrogen 750018], 0.1 mg/ml salmon sperm DNA [Agilent 201190], 1% SDS, 10% Dextran solution). Membrane was washed three times in 2× SSC/0.1% SDS (20× SSC: 3 M NaCl, 0.3 M Na Citrate, pH 7.0) and twice in 0.2× SSC/0.1% SDS, then exposed to a phosphor screen and developed using a phosphorimager (GE Healthcare Typhoon).

## Chromatin immunoprecipitation and library preparation

Chromatin immunoprecipitation (ChIP) was performed, with slight modification, as described by Thomas et al., 2015. Cells were first treated with 20 nM 4-OHT for 24 hr, then cross-linked in 1% methanol-free FA (Thermo Fisher 28908) for 10 min and quenched using 0.125 mM glycine. The cells were then rinsed twice in ice-cold 1× PBS, and lysed in formaldehyde lysis buffer (FALB: 50 mM HEPES pH 7.5, 140 mM NaCl, 1 mM EDTA, and 1% Triton X-100) + 1% SDS + PIC (Roche 05056489001). Sonication was performed in a BioRuptor (Diagenode, Denville, New Jersey) for 25 min, 30 s on/30 s off, and debris removed by centrifugation. To enable ChIP efficiency to be determined by qPCR, a 1:50 (2%) sample of chromatin was removed (input) prior to antibody addition.

For anti-HA, anti-rabbit IgG or anti-HCF1$_N$ ChIP, antibody was added to chromatin prepared from 12 × 10$^6$ Ramos cells, and samples rotated overnight at 4°C. Protein A Agarose (Roche 11134515001), blocked with 10 μg BSA, was added to each sample and rotated at 4°C for 2–4 hr. Washes were performed with Low Salt Wash Buffer (20 mM Tris pH 8.0, 150 mM NaCl, 2 mM EDTA, 1% Triton X-100), High Salt Wash Buffer (20 mM Tris pH 8.0, 500 mM NaCl, 2 mM EDTA, 1% Triton X-100), Lithium Chloride Wash Buffer (10 mM Tris pH 8.0, 250 mM LiCl, 1 mM EDTA, 1% Triton X-100), and twice with TE (10 mM Tris pH 8.0, 1 mM EDTA). Input and ChIP sample crosslinking were reversed at 65°C overnight in 50 μl TE + 200 mM NaCl + 0.1% SDS + 20–40 μg PK (Macherey-Nagel 740506).

For sequencing, each biological replicate was generated by combining two to three independent ChIPs (anti-IgG from CRE-ER$^{T2}$ parental cells, anti-HCF1$_N$ MYC-WT, 4A, or VP16 HBM cells, or anti-HA from CRE-ER$^{T2}$ parental cells, MYC-WT, 4A, or VP16 HBM cells)—performed using the same antibody on the same chromatin. DNA was extracted with phenol:chloroform:isoamyl alcohol, followed by ethanol precipitation in the presence of glycogen (Roche 10901393001) or GlycoBlue (Invitrogen AM9516). Libraries were prepared using NEBNext Ultra II DNA Library Prep Kit for Illumina (NEB E7645S) and NEBNext Multiplex Oligos for Illumina (NEB Set 1 E7335, Set 2 E7500S, NEB Unique Dual Index E6440S). Additional AMPure clean-ups at the start and the end of library preparation were included. Sequencing was carried out by Vanderbilt Technologies for Advanced Genomics using 75 bp paired-end sequencing on Illumina NextSeq 500 for anti-IgG and anti-HCF1$_N$ ChIPs or 150 bp paired-end sequencing on Illumina NovaSeq for anti-HA. The total number of sequencing reads for each replicate is shown in Supplementary file 2.

For qPCR, samples (either input or ChIP) were brought up to a final volume of 200 µl using TE. Each reaction was performed in a final volume of 15 µl, containing 2× SYBR FAST qPCR Master Mix (Kapa, Wilmington, Massachusetts, KK4602), 300 nM of each primer, and 2 µl of diluted sample. Three technical replicates were performed for each sample, and the mean Ct of these was used for calculating percent input. The mean Ct value for input was adjusted using the equation $Ct(input)-log_2(50)$. Percent input was then calculated using the equation $100 \times 2(adjustedCt - Ct(ChIP))$. Three biological replicates of ChIP-qPCR were performed using primers to amplify across the genes EXOSC5, UTP20, EIF2S1, POLR1A, EIF4G3, and HBB (β-Globin).

In vitro binding assays pSUMO-MYC WT-FLAG containing 6XHis- and FLAG-tagged full-length MYC (*Thomas et al., 2015*) was used for Q5 site-directed mutagenesis to substitute in the 4A (4A_F and 4A_R) and VP16 HBM (VP16 HBM_F and VP16 HBM_R) mutations. Rosetta cells Millipore 70954 were transformed with these plasmids, grown overnight, and induced the following day for 3 hr with 1 mM isopropyl β-d-1-thiogalactopyranoside (IPTG) at 30℃. Resulting cell pellets were resuspended in Buffer A (100 mM $NaH_2PO_4$, 10 mM Tris, 6 M GuHCl, 10 mM imidazole) + PIC (Roche 05056489001). Cells were sonicated 3× at 25% power for 10 s (Cole-Parmer GE 130PB-1), and debris removed by centrifugation; 150 µl bed volume Ni-NTA agarose (QIAGEN, Venlo, Netherlands, 30210) was added to the samples and rotated for 2 hr at 4℃. Beads were sequentially washed 1× with Buffer A, 1× with Buffer A/TI (1:3 Buffer A:Buffer TI), 1× with Buffer TI (25 mM Tris-HCl pH 6.8, 20 mM imidazole), and 1× with SUMO wash buffer (3 mM imidazole, 10% glycerol, 1× PBS, 2 mM DTT) + PIC (Roche 05056489001). Recombinant MYC was sequentially eluted twice from the beads using SUMO elution buffer (250 mM imidazole, 10% glycerol, 1× PBS, 2 mM DTT) + PIC (Roche 05056489001). The concentration of recombinant MYC was determined by resolving on a 10% SDS-PAGE gel alongside a BSA standard, and staining using Coomassie (50% methanol, 10% acetic acid, 0.1% w/v Coomassie Brilliant Blue).

HCF-1$_{VIC}$ (residues 1–380) from pCGT-HCF1$_{VIC}$ (*Thomas et al., 2016*) was cloned into pT7-IRES His-N (Takara 3290) using BamHI-HF (NEB R3136) and SalI-HF (NEB R3138), and Q5 site-directed mutagenesis was used to add an N-terminal T7 tag using the primers T7-HCF1_F and T7-HCF1_R. pT7-IRES His-T7-HCF-1$_{VIC}$ was in vitro transcribed/translated using the TnT Quick Coupled Transcription/Translation System (Promega, Madison, Wisconsin, L1171). Two milligrams of recombinant MYC and 12 µl of T7-HCF-1$_{VIC}$ were rotated overnight at 4℃ in Kischkel buffer + PIC (Roche 05056489001). Anti-FLAG M2 Affinity Gel (Sigma-Aldrich), blocked with 10 µg BSA, was added to each sample and rotated for 2 hr at 4℃. Beads were washed 4× in Kischkel buffer + 2 µg/ml Aprotinin (VWR, Radnor, Pennsylvania, 97062–752) + 1 µg/ml Pepstatin (VWR 97063–246) + 1 µg/ml Leupeptin (VWR 89146–578), and incubated in 1× Laemmli buffer for 5 min at 95℃.

## Electrophoretic mobility shift assays

MYC:MAX dimers were purified and prepared as described by *Farina et al., 2004*. pRSET-6XHis-MYC WT, a gift from Ernest Martinez, was used as a template for Q5 site directed mutagenesis to substitute in the 4A (4A_F and 4A_R) and VP16 HBM (VP16 HBM_F and VP16 HBM_R) mutations. The resulting plasmids, or pET-His-MAX, also from Ernest Martinez, were transformed into Rosetta cells (Millipore 70954), grown overnight, and induced the following day for 3 hr with 1 mM IPTG at 30℃. Resulting bacterial cell pellets were washed 1× with ice-cold wash buffer (10 mM Tris pH 7.9, 100 mM NaCl, 1 mM EDTA), then resuspended in lysis buffer (20 mM HEPES pH 7.9, 500 mM NaCl, 10% glycerol, 0.1% NP-40, 10 mM BME, 1 mM PMSF), and sonicated. Centrifugation was used to separate out the insoluble (pellet) and soluble (supernatant) fractions. For MAX, the recombinant protein is present in the supernatant and was then purified using Ni-NTA agarose (see below). For MYC samples, the supernatant was discarded, and the insoluble pellet was resuspended in E-buffer (50 mM HEPES pH 7.9, 5% glycerol, 1% NP-40, 10% Na-DOC, 0.5 mM BME), lysed using a Dounce homogenizer and B-pestle, and centrifuged. The pelleted inclusion bodies were lysed overnight in S-buffer (10 mM HEPES pH 7.9, 6 M GuHCl, 5 mM BME) by shaking at 25℃, and debris cleared by centrifugation.

Both the MAX supernatant and MYC supernatant from lysed inclusion bodies were adjusted to 5 mM imidazole. MYC and MAX were then bound to 75 µl bed volume Ni-NTA agarose (QIAGEN 30210). Successive washes were performed for 5 min each at 4℃: 3× with S-buffer + 5 mM imidazole, 3× with BC500 (20 mM Tris pH 7.9, 20% glycerol, 500 mM KCl, 0.05% NP-40, 10 mM BME, 0.2 mM PMSF) + 7 M urea + 5 mM imidazole, 1× with BC100 (20 mM Tris pH 7.9, 20% glycerol, 100

mM KCl, 0.05% NP-40, 10 mM BME, 0.2 mM PMSF) + 7 M urea + 15 mM imidazole, 1× with BC100 + 7 M urea + 30 mM imidazole. Elution was performed using BC100 + 7 M urea + 300 mM imidazole. Concentration of recombinant MYC and MAX was determined by running samples out on a 12% acrylamide gel alongside a BSA standard, and staining using Coomassie stain (50% methanol, 10% acetic acid, 0.1% w/v Coomassie Brilliant Blue).

For renaturation, 1.5 µg recombinant MAX was combined with 15 µg recombinant MYC (1:3 molar ratio), and brought up to a final volume of 150 µl using BC100 + 7 M urea. Dialysis was performed using a 'Slide-A-Lyzer' (Thermo Fisher 66383), with each dialysis step done for 2 hr stirring in the following solutions: BC500 + 0.1% NP-40 + 4 M urea at room temperature, BC500 + 0.1% NP-40 + 2 M urea at room temperature, BC500 + 0.1% NP-40 + 1 M urea at room temperature, BC500 + 0.1% NP-40 + 0.5 M urea at room temperature, BC500 at 4°C, and twice BC100 at 4°C. The product was centrifuged to remove debris, and BSA was added to a final concentration of 500 ng/µl.

Double-stranded labeled E-box probe (biotin group at the 3' end) and unlabeled competitors were prepared with dsDNA buffer (30 mM Tris pH 7.9, 200 mM KCl), and incubated at 95°C for 5 min. The E-box sequence used was 5'-GCTCAGGGACCACGTGGTCGGGGATC-3' and the mutant E-box sequence used was 5'-GCTCAGGGACCAGCTGGTCGGGGATC-3' (IDT). Double-stranded probe and the specific and non-specific competitors were prepared by combining 25 µM of each strand in dsDNA buffer (30 mM Tris pH 7.9, 200 mM KCl). For the probe, the forward strand carried a 3' biotin group; 20 fmol of labeled probe was bound to 0.55 pmol MYC:MAX or 0.06 pmol MAX:MAX dimers in the presence of 20 ng poly(dI-dC) (Thermo Fisher 20148E) in binding reaction buffer (15 mM Tris pH 7.9, 15% glycerol, 100 mM KCl, 0.15 mM EDTA, 0.075% NP-40, 7.5 mM BME, 375 ng/µl BSA) for 30 min at room temperature. For reactions involving unlabeled specific or nonspecific competitor, these were included in the binding reaction at a 100-fold excess over the biotinylated probe. EMSA gel loading solution (Thermo Fisher 20148K) was added to each sample and these were loaded onto a pre-run 6% polyacrylamide gel in 0.5× TBE (45 mM Tris, 45 mM boric acid, 1 mM EDTA). The gel was transferred to Hybond-N+ nylon membrane (GE Healthcare RPN303B) in 0.5× TBE for 30 min at 100 V. The remainder of the protocol was performed using LightShift Chemiluminescent EMSA Kit (Thermo Fisher 20148) according to manufacturer's instructions.

## RNA preparation, RT-qPCR, and RNA-Seq

Cell pellets were resuspended in 1 ml TRIzol (Invitrogen 15596026), and RNA was prepared according to the manufacturer's instructions. For switchable MYC allele Ramos cells, cells were treated with 20 nM 4-OHT for 24 hr and harvested. Prepared RNA was submitted to GENEWIZ (South Plainfield, NJ) for DNAse treatment, rRNA depletion, library preparation, and 150 bp paired-end sequencing on Illumina HiSeq. For untagged or FKBP^FV-HCF-1_N Ramos cells, cells were treated with DMSO or 500 nM dTAG-47 for 3 hr, and prepared RNA was DNAse treated prior to submission to Vanderbilt Technologies for Advanced Genomics for rRNA depletion, library preparation, and 150 bp paired-end sequencing on Illumina NovaSeq 6000. The total number of sequencing reads for each replicate is shown in *Supplementary file 2*.

For validation of RNA-seq by reverse transcriptase qPCR (RT-qPCR), RNA was prepared as above and 1 µg converted to cDNA using M-MLV reverse transcriptase (Promega M1701) in the presence of random hexamers (Invitrogen N8080127), RNase inhibitor (Thermo Fisher Scientific N8080119), and dNTPs (NEB N0446S). The resulting cDNA was brought up to a final volume of 160 µl using water. qPCR was performed in a final volume of 15 µl, containing 2× SYBR FAST qPCR Master Mix (Kapa), 300 nM of each primer, and 2 µl of diluted sample. Three technical replicates were performed for each sample, and the mean Ct of these was used for calculating FC. The mean Ct value for the gene of interest (GOI) was normalized to GAPDH (ΔCt) using the equation Ct(GOI)-Ct(GAPDH). For switchable MYC allele Ramos cells, ΔΔCt was calculated between treated (+4-OHT) and untreated (−4-OHT) cells. For FKBP^FV-HCF-1_N Ramos cells, ΔΔCt was calculated between dTAG-47-treated and DMSO-treated cells. FC was then calculated using the equation $2^{(-\Delta\Delta Ct)}$. Three biological replicates of RT-qPCR were performed. Primer sequences used are listed in *Supplementary file 1*.

## Next-generation sequencing analyses

After adapter trimming by Cutadapt (*Martin, 2011*), RNA-Seq reads were aligned to the hg19 genome using STAR (*Dobin et al., 2013*) and quantified by featureCounts (*Liao et al., 2014*). Differential analysis were performed by DESeq2 (*Love et al., 2014*), which estimated the log2 FCs, Wald test p-values, and adjusted p-value (false discovery rate, FDR) by the Benjamini–Hochberg procedure. The significantly changed genes were chosen with the criteria FDR < 0.05. ChIP-Seq reads were aligned to the hg19 genome using Bowtie2 (*Langmead et al., 2009*) after adapter trimming. Peaks were called by MACS2 (*Feng et al., 2012*) with a q-value of 0.01. ChIP read counts were calculated using DiffBind (*Stark and Brown, 2011*) and differential peaks were determined by DESeq2 (*Love et al., 2014*). Peaks were annotated using Homer command annotatePeaks, and enriched motifs were identified by Homer command findMotifsGenome (http://homer.ucsd.edu/homer/). All genomics data were deposited at GEO with the accession number GSE152385. Reviewers may access these data with the token 'enanoauyrpmfrmb'.

Details for the referenced MYC ChIP-Seq experiments from *Thomas et al., 2019* are available in *Tansey et al., 2019* (samples: GSM3593604–GSM3593606 and GSM3593616–3593618).

## Cell growth and glutamine deprivation assays

To generate a growth curve and determine doubling time of the MYC mutants, switchable *MYC* cells were treated with 20 nM 4-OHT for 16 hr, and then grown for a further 24 hr. Cells were then plated at a density of 20,000 cells/ml, and counted 3 and 6 days later. Growth rate (GR) was determined through the equation GR = $ln(N(7) - N(1))/144$, where $N(7)$ is the number of cells per milliliter on day 7, $N(1)$ is the number of cells per milliliter on day 1, and 144 is the number of hours that elapsed between the two measurements. Doubling time (DT) was then determined by the equation DT = $ln(2)/GR$. To measure the impact of HCF-1$_N$ degradation on cell growth, FKBP$^{FV}$-HCF-1$_N$ Ramos cells were plated at a density of 20,000 cells/ml with either DMSO or 500 nM dTAG-47. Cells were then counted every 24 hr for the following 4 days, without replacement of the compound or changing of the media.

To measure the impact of altering the MYC−HCF-1 interaction on cell growth, switchable MYC allele Ramos cell lines were treated with 20 nM 4-OHT for 2 hr to create a 50/50 mix of GFP-negative (WT) to GFP-positive (mutant) cells. Cells were sampled 24 hr later, and every 3 days following. Sampled cells were fixed in 1% FA in PBS for 10 min at room temperature, and the proportion of GFP-positive cells was determined using a BD LSR II flow cytometer at the Vanderbilt Flow Cytometry Shared Resource. To account for variation in the proportion of GFP-positive cells, each replicate was normalized to the proportion of GFP-positive cells at 24 hr post-treatment with 4-OHT.

To measure the impact of altering the MYC−HCF-1 interaction on glutamine dependence, switchable MYC allele Ramos cell lines were treated with 20 nM 4-OHT for 2 hr, and allowed to recover for 3 days. Cells were then split into RPMI 1640 without L-glutamine (Corning 15–040-CV), supplemented with 10% dialyzed FBS (Gemini Bio, West Sacramento, California, 100–108), and 1% P/S (Gibco 15140122), and grown for 16 hr with or without *Supplementary file 2* mM glutamine (Gibco 25030081). Glutamine was added back to the cells that were deprived, grown for 3 days, and fixed in 1% FA in PBS for 10 min at room temperature. The proportion of GFP-positive cells was determined using a BD LSR II flow cytometer, and normalized to the proportion of GFP-positive cells prior to being grown with or without supplemental glutamine. For the 4A and VP16 HBM cells, the proportion of GFP-positive cells was normalized to that in WT for glutamine supplementation (Gln+) or deprivation (Gln−).

## Metabolomics

### Sample preparation

Global, untargeted metabolomics was performed on switchable MYC allele Ramos cell lines treated with 20 nM 4-OHT for 24 hr. Individual cell pellet samples were lysed using 200 μl ice cold lysis buffer (1:1:2, acetonitrile:methanol:ammonium bicarbonate 0.1 M, pH 8.0, LC-MS grade) and sonicated using a probe tip sonicator, 10 pulses at 30% power, cooling down on ice between samples. A bicinchoninic acid protein assay was used to determine the protein concentration for individual samples, and adjusted to 200 μg total protein in 200 μl of lysis buffer. Isotopically labeled standard molecules, Phenylalanine-D8 and Biotin-D2, were added to each sample to assess sample extraction

quality. Samples were subjected to protein precipitation by addition of 800 µl of ice cold methanol (4× by volume), and incubated at −80°C overnight. Samples were centrifuged at 10,000 RPM for 10 min to eliminate precipitated proteins and supernatant(s) were transferred to a clean microcentrifuge tube and dried down *in vacuo*. Samples were stored at −80°C prior to LC-MS analysis.

## Global untargeted LC-MS/MS analysis

For mass spectrometry analysis, individual samples were reconstituted in 50 µl of appropriate reconstitution buffer (HILIC: acetonitrile/ H$_2$O, 90:10, v/v, RPLC: acetonitrile/H$_2$O with 0.1% formic acid, 3:97, v/v). Samples were vortexed well to solubilize the metabolites and cleared by centrifugation using a benchtop mini centrifuge to remove insoluble material. Quality control samples were prepared by pooling equal volumes from each sample. During final reconstitution, isotopically labeled standard molecules, Tryptophan-D3, Carnitine-D9, Valine-D8, and Inosine-4N15, were spiked into each sample to assess LC-MS instrument performance and ionization efficiency.

High resolution (HR) MS and data-dependent acquisition (MS/MS) analyses were performed on a high resolution Q-Exactive HF hybrid quadrupole-Orbitrap mass spectrometer (Thermo Fisher Scientific, Bremen, Germany) equipped with a Vanquish UHPLC binary system and autosampler (Thermo Fisher Scientific, Germany) at the Vanderbilt Center for Innovative Technology. For HILIC analysis metabolite extracts (10 µl injection volume) were separated on a SeQuant ZIC-HILIC 3.5 µm, 2.1 mm ×100 mm column (Millipore Corporation, Darmstadt, Germany) held at 40°C. Liquid chromatography was performed at a 200 µl/min using solvent A (5 mM ammonium formate in 90% H$_2$O, 10% acetonitrile) and solvent B (5 mM ammonium formate in 90% acetonitrile, 10% H$_2$O) with the following gradient: 95% B for 2 min, 95–40% B over 16 min, 40% B held 2 min, and 40–95% B over 15 min, 95% B held 10 min (gradient length: 45 min). For the RPLC analysis metabolite extracts (10 µl injection volume) were separated on a Hypersil Gold, 1.9 µm, 2.1 mm × 100 mm column (Thermo Fisher) held at 40°C. Liquid chromatography was performed at a 250 µl/min using solvent A (0.1% formic acid in H$_2$O) and solvent B (0.1% formic acid in acetonitrile) with the following gradient: 5% B for 1 min, 5–50% B over 9 min, 50–70% B over 5 min, 70–95% B over 5 min, 95% B held 2 min, and 95–5% B over 3 min, 5% B held 5 min (gradient length: 30 min).

Full MS analyses were acquired over a mass range of m/z 70–1050 using electrospray ionization positive mode. Full mass scan was used at a resolution of 120,000 with a scan rate of 3.5 Hz. The automatic gain control (AGC) target was set at $1 \times 10^6$ ions, and maximum ion injection time was at 100 ms. Source ionization parameters were optimized with the spray voltage at 3.0 kV, and other parameters were as follows: transfer temperature at 280°C; S-Lens level at 40; heater temperature at 325°C; Sheath gas at 40, Aux gas at 10, and sweep gas flow at 1.

Tandem mass spectra were acquired using a data-dependent scanning mode in which one full MS scan (m/z 70–1050) was followed by 2, 4 or 6 MS/MS scans. MS/MS scans were acquired in profile mode using an isolation width of 1.3 m/z, stepped collision energy (NCE 20, 40), and a dynamic exclusion of 6 s. MS/MS spectra were collected at a resolution of 15,000, with an AGC target set at $2 \times 10^5$ ions, and maximum ion injection time of 100 ms. The retention times and peak areas of the isotopically labeled standards were used to assess data quality.

## Metabolite data processing and analysis

LC-HR MS/MS raw data were imported, processed, normalized, and reviewed using Progenesis QI v.2.1 (Non-linear Dynamics, Newcastle, UK). All MS and MS/MS sample runs were aligned against a quality control (pooled) reference run, and peak picking was performed on individual aligned runs to create an aggregate data set. Unique ions (retention time and m/z pairs) were grouped (a sum of the abundances of unique ions) using both adduct and isotope deconvolutions to generate unique 'features' (retention time and m/z pairs) representative of unannotated metabolites. Data were normalized to all features using Progenesis QI. Compounds with <25% coefficient of variance (%CV) were retained for further analysis. Variance stabilized measurements achieved through log normalization were used with Progenesis QI to calculate p-values by one-way analysis of variance (ANOVA) test and adjusted p-values (Q-values). Significantly changed metabolites were chosen with the criteria Q-value <0.05 and |FC| > 1.5.

Tentative and putative identifications were determined within Progenesis QI using accurate mass measurements (<5 ppm error), isotope distribution similarity, and fragmentation spectrum matching

based on database searches against Human Metabolome Database (HMDB) (*Wishart et al., 2013*), METLIN (*Smith et al., 2005*), the National Institute of Standards and Technology (NIST) database (*Salvat et al., 2016*), and an in-house library. In these experiments, the level system for metabolite identification confidence was utilized (*Schrimpe-Rutledge et al., 2016*). Briefly, many annotations were considered to be tentative (level 3, L3) and/or putative (level 2, L2); in numerous circumstances a top candidate cannot be prioritized, thus annotations may represent families of molecules that cannot be distinguished. Data are available at the NIH Common Fund's National Metabolomics Data Repository (NMDR) Web site, the Metabolomics Workbench, https://www.metabolomicsworkbench. org where it has been assigned Study ID (ST001429).

## In vivo studies

### Tumor formation and maintenance assays

Six-week-old athymic nude mice (female *Foxn1*$^{nu/nu}$; Envigo, Indianapolis, IN) were injected subcutaneously into one flank with $10^7$ switched or unswitched WT, Δ264, 4A-1 or 4A-2 cells at the Thomas Jefferson Research Animals Shared Resource Core (IACUC protocol #01770). Mice were maintained in groups for all experiments, unless they were the last surviving member of a cohort. 4A-1 and 4A-2 carry the same 4A mutation, but were independent clones obtained from the same population. To facilitate lymphoma cell seeding, mice received whole-body irradiation (6 Gy) 24 hr prior to cell injection. For tumor formation studies, lymphoma cells were treated in vitro with 4-OHT for 24 hr to induce the switchable MYC cassette and expanded for 2 days prior to being injected into mice. For tumor maintenance studies, mice were injected with unswitched cells and allowed to form palpable tumors. Once tumors reached approximately 200 mm$^3$, mice received intraperitoneal injections of tamoxifen (2 mg in corn oil, Sigma-Aldrich T5648) once daily for three consecutive days to induce the switchable MYC cassette in vivo. Digital calipers were used to measure tumors and volumes calculated using the ellipsoid formula. Mice were sacrificed at humane endpoints based on tumor volume. Kaplan–Meier survival analyses were compared by log-rank tests to determine statistical significance. For lymphoma cell apoptosis evaluation, a cohort of mice with size-matched tumors prior to tamoxifen injection were sacrificed 48 and 96 hr following the first administration of tamoxifen. Flow cytometry (BD LSRII) at Thomas Jefferson University Flow Cytometry Shared Resource was used to measure fragmented (subG1) apoptotic DNA with propidium iodide (Sigma-Aldrich P4170), Annexin V/7AAD (BD Pharmingen 559763), and Caspase three activity (BD Pharmingen) in the lymphoma cells isolated from the tumors, as we previously reported (*Adams et al., 2017*) and as per manufacturer's protocols. Two-tailed t-tests were used to determine significance when comparing two groups. All mouse experiments were approved by the Institutional Animal Care and Use Committee at Thomas Jefferson University and complied with state and federal guidelines.

### Tumor gDNA analysis

To determine the proportion of cells that remain switched in the resulting tumors, gDNA was prepared from tumors using the PureLink Genomic DNA Mini Kit (Invitrogen K182002) according to manufacturer's instructions. As described in *Thomas et al., 2019*, gDNA from 0% switched (switchable MYC WT cells grown in puromycin) and 100% switched (permanently switched clonal cells) was also prepared in this manner for normalization. The resulting gDNA was diluted down to 50 ng/μl, so that 2 μl (100 ng) was loaded per well for qPCR. qPCR was performed using 2× SYBR FAST qPCR Master Mix (Kapa, Wilmington, Massachusetts, KK4602) in a Bio-Rad CFX96 Real-Time System. For each primer set (MYCP-4 and MYCP-5; SNHG15_F and SNHG15_R) and each independent tumor replicate, an average of three qPCR wells (technical replicates) was used. MYCP-4 and MYCP-5 primer set only amplifies gDNA from unswitched cells, whereas SNHG15_F and SNHG15_R amplifies gDNA from both unswitched and switched cells. For each gDNA sample, ΔCt was calculated as the difference between MYCP and SNHG15. gDNA from the 0% and 100% switched cells was used to calculate ΔΔCt, which was then used to normalize the ΔCt for the tumor gDNA to estimate the proportion of switched cells.

### Tumor RNA-Seq

Tumors from mice sacrificed 48 hr after the first tamoxifen administration were submitted to GENEWIZ for RNA extraction, DNAse treatment, rRNA depletion, library preparation, and 150 bp paired-

end sequencing on Illumina HiSeq. Four tumors for each WT, 4A-1, 4A-2, and Δ264 were submitted. The total number of sequencing reads for each replicate is shown in *Supplementary file 2*.

### Pathway and GO analysis, and figure generation

Classification of annotated metabolites was extracted from HMDB (*Wishart et al., 2018*) and LIPID MAPS (*Fahy et al., 2009*). Metabolite pathway analyses were performed using MetaboAnalyst 4.0 (*Chong et al., 2019*), and GO analyses using Metascape (*Zhou et al., 2019*) or DAVID (*Huang et al., 2009a*; *Huang et al., 2009b*). Chord diagrams were created using Circos (*Krzywinski et al., 2009*), pathways using Cytoscape (*Shannon et al., 2003*), and bubble plots using ggplot2 (*Wickham, 2016*).

### Statistical analysis and replicates for non-high throughput data

Unless otherwise stated, all experiments were conducted with at least three independent, biological replicates, and statistical tests were carried out using PRISM 8 (GraphPad).

Materials and resources mentioned here are available upon request.

## Acknowledgements

We thank Winship Herr for comments on the manuscript. For reagents we thank James Bradner, Feng Zhang, and Ernest Martinez. We thank the Vanderbilt Institute of Chemical Biology Synthesis Core for synthesis of dTAG-47. The VU Flow Cytometry Shared Resource is supported by the Vanderbilt Ingram Cancer Center (P30CA68485) and the Vanderbilt Digestive Disease Research Center (DK058404). VANTAGE is supported by the Vanderbilt Ingram Cancer Center, the Vanderbilt Digestive Disease Research Center, and the Vanderbilt Vision Center (P30EY08126). The Thomas Jefferson Flow Cytometry and Research Animals shared resource cores are supported by the Sidney Kimmel Cancer Center (NCI/NIH P30CA056036). This work was supported in part using the resources of the Center for Innovative Technology at Vanderbilt University. This work was supported by the Vanderbilt International Scholars Program (TMP), Robert J Kleberg, Jr., and Helen C Kleberg Foundation (WPT), Edward P Evans Foundation (WPT), the NCI/NIH (CA200709; WPT), the NCI/NIH (CA211305; LRT), the NCI/NIH (CA148950 CME), the Integrated Biological Systems Training in Oncology Training Program (T32 CA119925; AMW), the Rally Foundation for Childhood Cancer Research Fellowship (AMW), Open Hands Overflowing Hearts co-funded research fellowship (AMW), and the American Association for Cancer Research Basic Cancer Research Fellowship (AMW), the Herbert A Rosenthal, MD' 56 endowed professorship in Cancer Research (CME), the Steinfort family fund (CME), and the Sidney Kimmel Cancer Center/Thomas Jefferson University (CME, CMA).

## Additional information

### Funding

| Funder | Grant reference number | Author |
|---|---|---|
| Vanderbilt University | International Scholars Program | Tessa M Popay |
| Robert J. Kleberg, Jr. and Helen C. Kleberg Foundation | Targeting MYC | William P Tansey |
| Edward P. Evans Foundation | Grant in aid | William P Tansey |
| NIH | CA200709 | William P Tansey |
| NIH | CA211305 | Lance R Thomas |
| NIH | CA148950 | Christine Eischen |
| NIH | CA119925 | April M Weissmiller |
| Rally Foundation | Childhood Cancer Research Fellowship | April M Weissmiller |
| Rally Foundation | Open Hands Overflowing Hearts co-funded research | April M Weissmiller |

| | fellowship | |
| American Association for Cancer Research | Basic Cancer Research Fellowship | April M Weissmiller |
| Herbert A. Rosenthal, MD '56 endowed Professorship in Cancer Research | Professorship | Christine Eischen |
| Steinfort Family Fund | | Christine Eischen |

The funders had no role in study design, data collection and interpretation, or the decision to submit the work for publication.

### Author contributions

Tessa M Popay, Conceptualization, Data curation, Formal analysis, Investigation, Visualization, Methodology, Writing - original draft, Writing - review and editing; Jing Wang, Data curation, Formal analysis, Visualization, Writing - original draft, Writing - review and editing; Clare M Adams, Conceptualization, Investigation, Visualization, Writing - original draft, Writing - review and editing; Gregory Caleb Howard, Investigation, Visualization; Simona G Codreanu, Stacy D Sherrod, Resources, Investigation, Writing - original draft, Writing - review and editing; John A McLean, Resources, Funding acquisition, Methodology; Lance R Thomas, Conceptualization, Investigation, Methodology, Writing - original draft, Writing - review and editing; Shelly L Lorey, Supervision, Methodology, Writing - original draft, Writing - review and editing; Yuichi J Machida, Resources, Methodology, Writing - original draft, Writing - review and editing; April M Weissmiller, Conceptualization, Formal analysis, Validation, Investigation, Methodology, Writing - original draft, Writing - review and editing; Christine M Eischen, Conceptualization, Formal analysis, Supervision, Writing - original draft, Project administration, Writing - review and editing; Qi Liu, Conceptualization, Formal analysis, Supervision, Funding acquisition, Writing - original draft, Writing - review and editing; William P Tansey, Conceptualization, Data curation, Formal analysis, Supervision, Funding acquisition, Visualization, Writing - original draft, Project administration, Writing - review and editing

### Author ORCIDs

Tessa M Popay (iD) https://orcid.org/0000-0002-4694-8804
Stacy D Sherrod (iD) http://orcid.org/0000-0002-2346-230X
Christine M Eischen (iD) https://orcid.org/0000-0003-4618-8996
William P Tansey (iD) https://orcid.org/0000-0002-3900-0978

### Ethics

Animal experimentation: All mouse experiments were approved (IACUC protocol #01770) by the Institutional Animal Care and Use Committee at Thomas Jefferson University and complied with state and federal guidelines.

### Decision letter and Author response

Decision letter https://doi.org/10.7554/eLife.60191.sa1
Author response https://doi.org/10.7554/eLife.60191.sa2

## Additional files

### Supplementary files

• Supplementary file 1. Primer sequences.
• Supplementary file 2. Next-generation sequencing read counts.
• Transparent reporting form

### Data availability

All genomics data were deposited at GEO with the accession number GSE152385. Metabolomics data are available at the NIH Common Fund's National Metabolomics Data Repository (NMDR) Web

site, the Metabolomics Workbench, https://www.metabolomicsworkbench.org where it has been assigned Study ID (ST001429). Source data files have been provided for Figure 1, Figure 2, Figure 3, Figure 4, Figure 5, Figure 6 and Figure 7.

The following datasets were generated:

| Author(s) | Year | Dataset title | Dataset URL | Database and Identifier |
|---|---|---|---|---|
| Popay TM, Tansey WP, Sherrod SD, Codreanu SG, McLean JA | 2020 | MYC regulates ribosome biogenesis and mitochondrial gene expression programs through its interaction with Host Cell Factor-1 | https://doi.org/10.21228/M8WD7R | Metabolomics Workbench, 10.21228/M8WD7R |
| Popay TM, Tansey WP, Wang J, Liu Q | 2020 | MYC regulates ribosome biogenesis and mitochondrial gene expression programs through its interaction with Host Cell Factor-1 | https://www.ncbi.nlm.nih.gov/geo/query/acc.cgi?acc=GSE152385 | NCBI Gene Expression Omnibus, GSE152385 |

The following previously published dataset was used:

| Author(s) | Year | Dataset title | Dataset URL | Database and Identifier |
|---|---|---|---|---|
| Tansey WP, Thomas LR, Liu Q, Wang J | 2019 | Interaction with WDR5 recruits MYC to a small cohort of genes required for tumor onset and maintenance | https://www.ncbi.nlm.nih.gov/geo/query/acc.cgi?acc=GSE126207 | NCBI Gene Expression Omnibus, GSE126207 |

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
