## [Decision Letter]

**Acceptance summary:**

This manuscript is focused on the relevance of the interaction between MYC and HCF1. While it has been known for quite some time that MYC and HCF1 associate in mammalian cells and in *Drosophila*, there is a lack of compelling evidence relating to the extent or the importance of this association, in particular its potential importance in MYC oncogenic activity. The present paper attempts to address these issues. The authors identify a short sequence (they term HBM) in MycBox4 that mediates association with HCF1 and furthermore devise a clever strategy to inducibly mutate this sequence in Ramos cells. By manipulating expression of the HBM mutant MYC as well as HCF1 itself they are able to characterize the effects on altering the MYC-HCF1 interaction at the level of proliferation, global gene expression, and tumor growth. Moreover, they examine genomic occupancy and binding overlap of wildtype MYC and HCF1. The results taken together make a good case for important regulatory and functional consequences of the interaction on a distinct subset of genes that are predominantly involved in ribosome biogenesis and mitochondrial and amino acid metabolism; often considered to be "core" activities of MYC.

**Decision letter after peer review:**

Thank you for submitting your article "MYC regulates ribosome biogenesis and mitochondrial gene expression programs through interaction with Host Cell Factor-1" for consideration by *eLife*. Your article has been reviewed by three peer reviewers, including Martin Eilers as the Reviewing Editor and Reviewer #1, and the evaluation has been overseen by Maureen Murphy as the Senior Editor. The following individual involved in review of your submission has agreed to reveal their identity: Elmar Wolf (Reviewer #2).

The reviewers have discussed the reviews with one another and the Reviewing Editor has drafted this decision to help you prepare a revised submission.

Summary:

All reviewers agree this is a very interesting paper that validates the interaction between MYC and HCF1 as biologically relevant in a tumor model and provides a lot of interesting data on what the interaction may do. At the same time, all three reviewers make suggestions as to how to improve the impact of the paper and raise several new questions.

Reviewer #1:

This is a comprehensive and technically well-executed manuscript that validates an interaction between MYC and HCF1 that has been previously demonstrated by the authors. The data are interesting with respect to effects on gene expression, while the impact on cellular metabolism remain less clear. Potentially the most surprising aspect is that – while MYC and HCF1 co-bind to a number of promoters – this appears to largely independent of the effect on gene expression. Collectively, the data clearly validate the interaction as being biologically relevant but leave at this reviewer without a clear take-home message. The strongest data are the ones that document a dependence of established tumors on the MYC/HCF1 interaction, which in my view clearly warrant publication, I would have a number of specific comments.

1) The effect of the switch between mutants on cell growth is measured with an ultrasensitive cell competition; this is relevant, since there is no effect on the relative distribution on cell cycle phases, suggesting that the "real" effect may be subtle. The authors need to show straightforward growth curves and then use this together with the cell cycle phases distribution to calculate the length of each cell cycle phase.

2) The Z-normalization needs to be removed from changes in metabolite levels and, as presented, the data in Figure 2—figure supplement 1 are not informative, they are simply blue and red tiles.

3) It is not clear to this reviewer what the authors can conclude from the changes in steady-state levels of metabolites. Also, parameters like "pathway impact" seem very indirect conclusions. In the absence of flux data that they show that any of these pathways is affected, I would suggest to shorten or delete the metabolic data.

4) There must be a better way to represent the data shown in Figure 3D and E. I think the authors will agree that – just to give one example – dark blue lines between unnamed genes in the category "ribosome biogenesis" to unreadable individual genes are simply not informative

5) It is not clear to this reviewer why the authors Z-normalize their gene expression data and then do Go-term enrichment analyses. Why not simply rung GSEA analyses on them? This would allow a much simpler comparison with other datasets. for example, the effects of the mutations on the MYC (and other) Hallmark gene sets should be shown.

6) Figure 5G should be shown for an equal number of random non co-bound genes, which have the same expression levels as the median of the co-bound genes.

7) The tumor data for the VP16 replacement need to be shown. It may well be that the MYC/HCF1 interaction is attenuated in wildtype MYC to prevent excessive MYC function.

8) As the authors know very well, there is a view of MYC as a global regulator of RNAPII function. It is possible, therefore, that they overlook a global change in RNAPII function, and this may resolve some of the puzzles here. They should show one RNAPII ChIPseq (total, pS2, pS5) experiment that addresses the issue and stratify this for the different sets of genes they are interested in.

Reviewer #2:

In the manuscript "MYC regulates ribosome biogenesis and mitochondrial gene expression programs through interaction with Host Cell Factor-1" Tessa Popay and her colleagues from the Tansey lab describe that regulation of a distinct group of MYC target genes depends on interaction with HCF-1. In an elegant set of experiments, the authors constructed both, loss- and gain of function mutants of the MYC/HCF-1 interaction and characterized them in multiple assays.

1) Alanine substitutions in MBIV ("4A") abolish interaction to HCF-1. Other substitutions increase interaction (VP16HBM).

2) Switchable expression to 4A decreases RAMOS long term growth in vitro. VP16HBM increases.

3) 4A and VP16HBM changes amino acid metabolism.

4) Gene expression analysis demonstrated downregulation of RIBI and mitochondrial matrix genes after switch to 4A. Opposite observation after switch to VP16HBM.

5) Acute depletion of HCF-1: Downregulation od RIBI genes.

6) MYC and HCF-1 bind to a small common subset of target genes, which is different to MYC/WDR5 target genes but functionally similar.

7) Switch to 4a compromises tumor engraftment and inhibits tumorigenesis.

The topic of the paper is of general interest to scientists working on tumor biology and the function of MYC. The data is overall convincing and the paper is very nicely written. I support publication in *eLife* after few comments could be addressed:

1) The authors claim that the mutations in MBIV affect the "direct" interaction between MYC and HCF-1. This conclusion is supported by recombinantly expressed MYC and IVT HCF-1. I think, one cannot really conclude direct interaction by this assay, as the reticulocyte extract contains all other mammalian proteins. The authors should show direct interaction by other means (i.e. purified HCF-1) or tone down their statements (see also in Discussion). One would at least expect a Coomassie gel for MYC.

2) The authors should think of assays, which demonstrate that 4A/ VP16HBM MYC mutants are normal with the exception if binding HCF-1 (IF, Reporter assays, ChIP,.…). I think, this is important.

3) I like the heatmap shown in Figure 3—figure supplement 3C summarizing the RNA-seq results (maybe show in the main Figure? Maybe instead of the radar plots?). How do the authors interpret that about one third of all genes, which are regulated in respect to wildtype MYC, are regulated in the same direction? Interaction of MBIV to other binding partners? Maybe the authors want to discuss potential explanations?

4) The degron system seems to work very well – congratulations! Isn´t that an ideal system to support the authors conclusion "gene expression changes that result directly from the MYC-HCF-1 interaction occur through a mechanism that is independent of their recruitment to chromatin.", by doing MYC-ChIP-seqs after dTAG? That experiment is, however, not essential in my view.

Reviewer #3:

This highly interesting manuscript is focused on the relevance of the interaction between MYC and HCF1. While it has been known for quite some time that MYC and HCF1 associate in mammalian cells and in *Drosophila*, there is a lack of compelling evidence relating to the extent or the importance of this association, in particular its potential importance in MYC oncogenic activity. The present paper attempts to address these issues. The authors identify a short sequence (they term HBM) in MycBox4 that mediates association with HCF1 and furthermore devise a clever strategy to inducibly mutate this sequence in Ramos cells. By manipulating expression of the HBM mutant MYC as well as HCF1 itself they are able to characterize the effects on altering the MYC-HCF1 interaction at the level of proliferation, global gene expression, and tumor growth. Moreover, they examine genomic occupancy and binding overlap of wildtype MYC and HCF1. The results taken together make a good case for important regulatory and functional consequences of the interaction on a distinct subset of genes that are predominantly involved in ribosome biogenesis and mitochondrial and amino acid metabolism; often considered to be "core" activities of MYC. The authors state that these findings may provide leverage for inhibiting MYC. Overall, this is an impressive paper – the functional and biochemical data are extensive and convincing: the usage of CRISPR knock-ins is a particularly elegant and effective way to test the importance of the MYC-HCF1 interactions. In addition, the paper very well written and easy to follow.

Essential revisions:

One surprising finding of this study that is not explored or discussed in sufficient detail is that gene regulation by MYC-HCF1 occurs through a "co-recruitment independent mechanism". This conclusion is derived from their data that, although the MYC 4A HBM mutant no longer interacts with HCF1, both MYC and HCF1 binding to target genes is not appreciably affected. This was determined using ChIP-qPCR on a small number of targets (Figure 5I, J). This leaves the reader with no clear idea of the basis for the actual mechanism of cooperation and the authors offer scant guidance. It is certainly conceivable that MYC and HCF1 bind independently to targets and the HBM serves to stabilize or increase the affinity of interaction, modulate the relative positions of the bound proteins, or determine inclusion in a condensate. Importantly aside from the ChIP qPCR of a few targets, global binding data in (i) the presence of the MYC 4A mutant and (ii) the absence of HCF1 (dTAG) are not provided. It would be of considerably strengthen this paper to include analysis of gene tracks of MYC4A in presence and absence of HCF to determine whether there are shifts in the extent of binding or positioning of MYC and/or HCF peaks, changes in PolII distribution, and epigenetic alterations. Another explanation is that the MYC HBM sequence actually serves as a binding site for other co-factors, in addition to HCF1, which could impact MYC activity (see point 2 below).

1) Does the 4A mutation disrupt any other known interactions of MYC with other proteins? Carrying out a proteomic screen using WT vs 4A could identify other candidates.

2) All the experiments appear to be done using a single (EBV negative) BL line: Ramos. How broadly applicable is this interaction in terms of other lymphoma lines or tumor types?

3) What happens in the context of endogenous MYC? Is the HCF-MYC interaction still critical e.g. in normal B cells. This is particularly important to assess from a translational standpoint.

4) Are there documented mutations in patients in the MBIV HBM region that could potentially be gain-of-function (i.e. similar to the VP16 HBM)?

5) Figure 5: Was there a significant enrichment seen for canonical E-boxes? The NRF and Sp1 motifs are not particularly specific to MYC and I do not think that these sequences were found to bind MYC-MAX in the Blackwell et al. paper cited. Are the binding sites identified actually specific for dual binding by MYC and HCF1. In Figure 6, there is substantial overlap between gene expression changes in the tumor engraftment experiment when compared to the in vitro data. However, there are a lot of changes seen that are absent in the cell line. Is there a GO analysis for the in vivo specific signatures?

---

## [Author Response]

Reviewer #1:This is a comprehensive and technically well-executed manuscript that validates an interaction between MYC and HCF1 that has been previously demonstrated by the authors. The data are interesting with respect to effects on gene expression, while the impact on cellular metabolism remain less clear. Potentially the most surprising aspect is that – while MYC and HCF1 co-bind to a number of promoters – this appears to largely independent of the effect on gene expression. Collectively, the data clearly validate the interaction as being biologically relevant but leave at this reviewer without a clear take-home message. The strongest data are the ones that document a dependence of established tumors on the MYC/HCF1 interaction, which in my view clearly warrant publication, I would have a number of specific comments.1) The effect of the switch between mutants on cell growth is measured with an ultrasensitive cell competition; this is relevant, since there is no effect on the relative distribution on cell cycle phases, suggesting that the "real" effect may be subtle. The authors need to show straightforward growth curves and then use this together with the cell cycle phases distribution to calculate the length of each cell cycle phase.

I disagree with the reviewer's contention that the experiment in Figure 1H represents an "ultrasensitive" competition assay. At day zero, the switched and unswitched cells are present in a 50:50 ratio that is same for the WT, 4A, and VP16 HBM cells. I do agree, however, that any competition assay will give an exaggerated impression of differences, compared to stand alone studies of each of the individual experimental arms. We now include, therefore, the requested straightforward growth curves—both as raw cell numbers (Figure 1—figure supplement 2C) and as calculated doubling times (Figure 1—figure supplement 2D). These new data clearly show that 4A cells double more slowly than WT, whereas VP16 HBM cells double more quickly. The difference is two to three hours in either direction. I do not think it is prudent to extrapolate these differences out to specific cell cycle phases, because this would be highly speculative and would add little to the story. The important point we make is that the growth differences between the WT, 4A, and VP16 HBM cells are real—and quantified—and we thank the reviewer for prompting us to include this experiment.

2) The Z-normalization needs to be removed from changes in metabolite levels and, as presented, the data in Figure 2—figure supplement 1 are not informative, they are simply blue and red tiles.

The heatmaps in Figure 2—figure supplement 1 are shown with Z-normalization as a way to represent the consistency among replicates. We have added statements to the text indicating this purpose. We have also included, with each of these heatmaps, a representation of log_2_FC so that the magnitude of the changes associated with the 4A and VP16 HBM mutants can be assessed.

3) It is not clear to this reviewer what the authors can conclude from the changes in steady-state levels of metabolites. Also, parameters like "pathway impact" seem very indirect conclusions. In the absence of flux data that they show that any of these pathways is affected, I would suggest to shorten or delete the metabolic data.

We have taken the reviewer's advice and shortened this section of the manuscript, including removal of the "pathway impact" figure and its associated text. We do feel that the connections to steady state amino acid levels are worth reporting, so we have revised this section to focus specifically on how these are altered in response to the 4A and VP16 HBM mutants.

4) There must be a better way to represent the data shown in Figure 3D and E. I think the authors will agree that – just to give one example – dark blue lines between unnamed genes in the category "ribosome biogenesis" to unreadable individual genes are simply not informative

In response to this comment, we have replaced the Circos plots in Figure 3 with bubble plots that represent GO categories, gene counts, and significance. This is a much more informative way of presenting the data. There is value to the Circos plots, however, because they allow individual genes in the categories to be represented. We have now included high resolution versions of these plots in new Figure 2—figure supplement 2; gene names can be visualized by zooming in. We would like to include the Circos plots as part of this story but if the reviewer feels strongly that these should be removed entirely, we will oblige.

5) It is not clear to this reviewer why the authors Z-normalize their gene expression data and then do Go-term enrichment analyses. Why not simply rung GSEA analyses on them? This would allow a much simpler comparison with other datasets. for example, the effects of the mutations on the MYC (and other) Hallmark gene sets should be shown.

I apologize for the confusion regarding Z-normalization. We do not Z-normalize gene expression data and then perform GO term enrichment analysis; we only Z-normalize for the heatmaps so that we can show variation among biological replicates. We have revised the text to make this point clear. I do not think there is anything wrong with using the GO term enrichment analysis here in Figure 3B, especially as we back up our conclusions by performing GO analysis on the specific gene sets that are regulated in the same, or opposing, ways by the 4A and VP16 HBM mutants (Figure 3E–F).

While these RNA-Seq data are being discussed, I point out that we added a new piece of validation data (not requested) in Figure 3—figure supplement 1C demonstrating that select gene expression changes we observe in the RNA-Seq are a bonafide consequence of switching (*i.e.,* responsive to 4-OHT).

6) Figure 5G should be shown for an equal number of random non co-bound genes, which have the same expression levels as the median of the co-bound genes.

Excellent recommendation. We have performed this analysis and it is now included as Figure 5I in the revised manuscript. The analysis clearly shows that MYC–HCF-1 co-bound genes are significantly more affected by the 4A or VP16 HBM mutants than equivalently expressed genes that are not cobound.

7) The tumor data for the VP16 replacement need to be shown. It may well be that the MYC/HCF1 interaction is attenuated in wildtype MYC to prevent excessive MYC function.

As we note in the manuscript, the Ramos model is already very aggressive (see the tumor growth curves in Figure 7—figure supplement 1B), and it is not clear that we would be able to visualize an increase in tumor growth under these circumstances. We plan to test this mutant in other in vivo models in the future, particularly those with a slower rate of progression, so that we can test this idea that interaction with HCF is tempered to prevent excessive MYC function.

8) As the authors know very well, there is a view of MYC as a global regulator of RNAPII function. It is possible, therefore, that they overlook a global change in RNAPII function, and this may resolve some of the puzzles here. They should show one RNAPII ChIPseq (total, pS2, pS5) experiment that addresses the issue and stratify this for the different sets of genes they are interested in.

We have struggled with this comment, in large part because it is not at all clear what "puzzles" are being referred to here. I would argue that we have gone above and beyond what is typically done in terms of identifying these ribosome biogenesis and mitochondrial matrix genes as MYC-HCF-1 targets. We have a loss-of-function MYC mutant, a gain-of-function MYC mutant, HCF-1 degradation, and location on chromatin; all of which point to a very consistent set of genes. I think it could be interesting to look at the MYC–HCF-1 interaction and whether it influences the global functioning of MYC in the future, but I think that needs more than just an RNAPII ChIP and clearly extends beyond the scope of the current story.

Reviewer #2:In the manuscript "MYC regulates ribosome biogenesis and mitochondrial gene expression programs through interaction with Host Cell Factor-1" Tessa Popay and her colleagues from the Tansey lab describe that regulation of a distinct group of MYC target genes depends on interaction with HCF-1. In an elegant set of experiments, the authors constructed both, loss- and gain of function mutants of the MYC/HCF-1 interaction and characterized them in multiple assays.1) Alanine substitutions in MBIV ("4A") abolish interaction to HCF-1. Other substitutions increase interaction (VP16HBM).2) Switchable expression to 4A decreases RAMOS long term growth in vitro. VP16HBM increases.3) 4A and VP16HBM changes amino acid metabolism.4) Gene expression analysis demonstrated downregulation of RIBI and mitochondrial matrix genes after switch to 4A. Opposite observation after switch to VP16HBM.5) Acute depletion of HCF-1: Downregulation od RIBI genes.6) MYC and HCF-1 bind to a small common subset of target genes, which is different to MYC/WDR5 target genes but functionally similar.7) Switch to 4a compromises tumor engraftment and inhibits tumorigenesis.The topic of the paper is of general interest to scientists working on tumor biology and the function of MYC. The data is overall convincing and the paper is very nicely written. I support publication in eLife after few comments could be addressed:1) The authors claim that the mutations in MBIV affect the "direct" interaction between MYC and HCF-1. This conclusion is supported by recombinantly expressed MYC and IVT HCF-1. I think, one cannot really conclude direct interaction by this assay, as the reticulocyte extract contains all other mammalian proteins. The authors should show direct interaction by other means (i.e. purified HCF-1) or tone down their statements (see also in discussion). One would at least expect a Coomassie gel for MYC.

Excellent point. In response to this comment, we removed any mention of the interaction between MYC and HCF-1 being "direct" and added the Coomassie gel for recombinant MYC proteins as Figure 1—figure supplement 1A.

2) The authors should think of assays, which demonstrate that 4A/ VP16HBM MYC mutants are normal with the exception if binding HCF-1 (IF, Reporter assays, ChIP,.…). I think, this is important.

In response to this comment, we performed three additional sets of experiments, and include the new data in the revised manuscript. First, we carried out cellular fractionation assays (Figure 1—figure supplement 2A) and showed that both the A4 and VP16 HBM mutants reside within the nuclear fraction (P3), similar to wild-type MYC. Second, we performed immunofluorescence studies (Figure 1—figure supplement 2B) and showed that the A4 and VP16 HBM mutants are predominantly nuclear, again similar to wild-type MYC. Third, we performed ChIP-seq analysis (Figure 6 and Figure 6—figure supplement 1) and found that the localization of MYC on chromatin is largely unaffected by the 4A and VP16 HBM mutations; there are some changes, as we report, but these changes are subtle and trend in the same direction for the 4A and VP16 HBM mutants. In addition, and as included in the original submission, we show that the A4 and VP16 HBM MYC mutants retain DNA-binding capability in vitro (Figure 6—figure supplement 1), as well as the ability to interact with WDR5 (Figure 1D and Figure 1E). Finally, I point out that, although we cannot exclude the possibility that mutations in the HBM disrupt some unknown interaction partner or function of MYC, our focus on reciprocal changes emerging from parallel loss and gain of function approaches ameliorates many of the concerns that arise when a purely loss of function strategy is employed.

3) I like the heatmap shown in Figure 3—figure supplement 3C summarizing the RNA-seq results (maybe show in the main Figure? Maybe instead of the radar plots?). How do the authors interpret that about one third of all genes, which are regulated in respect to wildtype MYC, are regulated in the same direction? Interaction of MBIV to other binding partners? Maybe the authors want to discuss potential explanations?

We have now moved this heatmap into Figure 3, as requested. In response to reviewer #1 comment # 4 we also moved the Circos (radar) plots out of the main figure and into a new Supplemental figure. Our speculation for the common gene expression changes shared between the 4A and VP16 HBM mutants is that they both disrupt a function of MYC that is distinct from HCF-1. We have added text to the Discussion, as recommended by the reviewer, to make this point clear. We are grateful to the reviewer for prompting us to discuss this point in the manuscript, as it also strengthens our argument that the dual loss-of-function and gain-of-function approach we have taken allows us to pinpoint genes controlled via the MYC–HCF-1 interaction.

4) The degron system seems to work very well – congratulations! Isn´t that an ideal system to support the authors conclusion "gene expression changes that result directly from the MYC-HCF-1 interaction occur through a mechanism that is independent of their recruitment to chromatin.", by doing MYC-ChIP-seqs after dTAG? That experiment is, however, not essential in my view.

Thank you. The addition of the degron system for HCF-1 definitely strengthens the story, particularly in terms of the connections between MYC, HCF-1, and RiBi and mitochondrial gene expression programs. I am not convinced, however, that it would add much to the mechanism, beyond what we can deduce from the 4A and VP16 HBM mutants. And in fact, it could be misleading. It is entirely possible, for example, that loss of HCF-1 will alter the chromatin state of associated loci, influencing MYC recruitment in a very indirect way. In my opinion, the HBM point mutants are a much better way of asking whether contact between MYC and HCF-1 influences the recruitment of either protein to chromatin.

I realize, however, that we did not go far enough in our original studies to support our broad statements about a recruitment-independent mechanism, largely because we only looked at binding of MYC and HCF-1 to a handful of genes. We therefore went back and performed an entirely new set of MYC and HCF-1 ChIP-seq experiments in the WT, 4A, and VP16 HBM cells. The resulting data, which is now included in Figure 6 and Figure 6—figure supplement 1, shows that global chromatin binding by MYC and HCF-1 are largely unaffected by the 4A and VP16 HBM mutations; as stated earlier, there are some subtle changes for MYC, as we report in the manuscript, but these changes trend in the same direction for the 4A and VP16 HBM mutants. With these new data in hand, we can be confident that we have not missed critical genes where co-dependent recruitment oh MYC and HCF-1 occurs.

Reviewer #3:This highly interesting manuscript is focused on the relevance of the interaction between MYC and HCF1. While it has been known for quite some time that MYC and HCF1 associate in mammalian cells and in *Drosophila*, there is a lack of compelling evidence relating to the extent or the importance of this association, in particular its potential importance in MYC oncogenic activity. The present paper attempts to address these issues. The authors identify a short sequence (they term HBM) in MycBox4 that mediates association with HCF1 and furthermore devise a clever strategy to inducibly mutate this sequence in Ramos cells. By manipulating expression of the HBM mutant MYC as well as HCF1 itself they are able to characterize the effects on altering the MYC-HCF1 interaction at the level of proliferation, global gene expression, and tumor growth. Moreover, they examine genomic occupancy and binding overlap of wildtype MYC and HCF1. The results taken together make a good case for important regulatory and functional consequences of the interaction on a distinct subset of genes that are predominantly involved in ribosome biogenesis and mitochondrial and amino acid metabolism; often considered to be "core" activities of MYC. The authors state that these findings may provide leverage for inhibiting MYC. Overall, this is an impressive paper – the functional and biochemical data are extensive and convincing: the usage of CRISPR knock-ins is a particularly elegant and effective way to test the importance of the MYC-HCF1 interactions. In addition, the paper very well written and easy to follow.Essential revisions:One surprising finding of this study that is not explored or discussed in sufficient detail is that gene regulation by MYC-HCF1 occurs through a "co-recruitment independent mechanism". This conclusion is derived from their data that, although the MYC 4A HBM mutant no longer interacts with HCF1, both MYC and HCF1 binding to target genes is not appreciably affected. This was determined using ChIP-qPCR on a small number of targets (Figure 5 I, J). This leaves the reader with no clear idea of the basis for the actual mechanism of cooperation and the authors offer scant guidance. It is certainly conceivable that MYC and HCF1 bind independently to targets and the HBM serves to stabilize or increase the affinity of interaction, modulate the relative positions of the bound proteins, or determine inclusion in a condensate. Importantly aside from the ChIP qPCR of a few targets, global binding data in (i) the presence of the MYC 4A mutant and (ii) the absence of HCF1 (dTAG) are not provided. It would be of considerably strengthen this paper to include analysis of gene tracks of MYC4A in presence and absence of HCF to determine whether there are shifts in the extent of binding or positioning of MYC and/or HCF peaks, changes in PolII distribution, and epigenetic alterations. Another explanation is that the MYC HBM sequence actually serves as a binding site for other co-factors, in addition to HCF1, which could impact MYC activity (see point 2 below).

We appreciate the reviewer's interest in knowing the mechanism of action through which the MYC– HCF-1 interaction promotes transcription of direct target genes. But the reality is that, if one excludes co-recruitment as a mechanism, there are very few, if any, solid leads to pursue. It could take many years to figure this out, and there is no clear experimental trajectory that we can take (at the moment) to be sure to get us there. What we can do, however, and as the reviewer suggests, is fortify our conclusion that the mechanism is indeed co-recruitment independent; which we have now done.

In response to this comment, we performed ChIP-seq for MYC and for HCF-1 in the WT, 4A, and VP16 HBM cells after switching. These data (now included as a new Figure 6 and Figure 6—figure supplement 1) clearly show that recruitment of HCF-1 to chromatin is not impacted by the 4A or VP16 HBM mutations. They also show that recruitment of MYC to chromatin is largely unaffected by these mutations. As reported in the manuscript, we do see some changes in the binding of MYC to chromatin when the HBM is mutated, but these are subtle, and trend in the same direction for both the 4A and VP16 mutants—excluding the MYC–HCF-1 interaction as being responsible. We thank the reviewer for challenging us to perform these experiments because they now provide a much more solid foundation for our statements that the mechanism through which MYC and HCF-1 function to control transcription is *not* at the level of co-recruitment.

Additionally, we have included a statement in the Discussion that this area needs further investigation. We also developed the discussion of how a similar "recruitment independent" precedent exists for HCF-1, and offer one possible scenario for how this could occur.

1) Does the 4A mutation disrupt any other known interactions of MYC with other proteins? Carrying out a proteomic screen using WT vs 4A could identify other candidates.

This comment is conceptually similar to that laid out in reviewer #2 comment #2. It is possible that the 4A mutation disrupts interaction of MYC with multiple factors besides HCF-1. But it is precisely this concept that led us to implement a combined loss-of-function and gain-of-function approach in this study, and focusing on reciprocal changes that occur between the 4A and VP16 HBM mutants. This concept also underlies our use of HCF-1 depletion as a way of further anchoring our findings to the MYC–HCF-1 connection. If another factor is involved, it would have to be sensitive the 4A and VP16 mutations in exactly the same reciprocal way as HCF-1, and its functioning would need to be sensitive to HCF-1 depletion. Again, we cannot formally exclude this possibility, but our comprehensive approach meets—and I would argue exceeds—the contemporary standard for validation of MYC co-factor.

2) All the experiments appear to be done using a single (EBV negative) BL line: Ramos. How broadly applicable is this interaction in terms of other lymphoma lines or tumor types?

That is correct; all of our experiments have been done in Ramos cells. Given all the work involved, it is not feasible to extend to other cell or tumor types in this study. But the reviewer makes an important point—we do not know if this interaction will extend to other tumor types. To address this comment, we have added a statement to the beginning of the introduction that makes this point clear.

3) What happens in the context of endogenous MYC? Is the HCF-MYC interaction still critical e.g. in normal B cells. This is particularly important to assess from a translational standpoint.

We do not know. I agree this is important from a translational standpoint. But looking in normal Bcells, or indeed in other tumor types as mentioned above, clearly is beyond the scope of the present study. I note that we refer specifically to this point in both the first and the last paragraph of the Discussion and state that we do not know if a therapeutic window could be achieved if MYC is targeted via the HCF-1 interaction.

4) Are there documented mutations in patients in the MBIV HBM region that could potentially be gain-of-function (i.e. similar to the VP16 HBM)?

Yes. There are three examples listed in the COSMIC database in which the HBM of MYC is mutated. Interestingly, all three of these convert the "QHYN" of MYC to "EHNY", which perfectly matches the high affinity HBM consensus. We now discuss this point in the last paragraph of the Discussion. We thank the reviewer for asking this provocative question.

5) Figure 5: Was there a significant enrichment seen for canonical E-boxes? The NRF and Sp1 motifs are not particularly specific to MYC and I do not think that these sequences were found to bind MYC-MAX in the Blackwell et al. paper cited. Are the binding sites identified actually specific for dual binding by MYC and HCF1. In Figure 6, there is substantial overlap between gene expression changes in the tumor engraftment experiment when compared to the in vitro data. However, there are a lot of changes seen that are absent in the cell line. Is there a GO analysis for the in vivo specific signatures?

As discussed earlier, we realize that we placed too much emphasis on the NRF-1 motif in the original version of the manuscript. We were simply reporting that it is enriched in our motif analysis of HCF-1 binding sites and that it is likely an E-box variant—which makes sense given that almost all the HCF-1 peaks are co-bound by MYC. Delving deeper into the significance of this variant or the quality of Eboxes in general in not warranted, especially now the we have ChIP-seq data demonstrating that MYC and HCF-1 bind chromatin independent of their ability to interact with one another. We went back and looked carefully at the Blackwell et al., paper and we were correct in our statement that the CATGCG motif is a MYC:MAX binding site. In that paper, they present both gel-shift analysis, as well as methylation interference analysis, which clearly shows binding to this site in vitro (the relevant probe is "M18" in Figures 2 and 3). In response to this comment, we have revised and simplified the text to avoid giving the wrong impression, and we have cited an additional three papers that establish the CATGCG motif as an authentic MYC binding motif.

In response to the question of whether there is a GO analysis for the in vivo specific signatures, these have now been added as Figure 7—figure supplement 1J and 1K.